# Prolonged Immobilization Exacerbates the Loss of Muscle Mass and Function Induced by Cancer-Associated Cachexia through Enhanced Proteolysis in Mice

**DOI:** 10.3390/ijms21218167

**Published:** 2020-10-31

**Authors:** Laura Mañas-García, Antonio Penedo-Vázquez, Adrián López-Postigo, Jorieke Deschrevel, Xavier Durán, Esther Barreiro

**Affiliations:** 1Pulmonology Department-Muscle Wasting and Cachexia in Chronic Respiratory Diseases and Lung Cancer Research Group, IMIM-Hospital del Mar, Parc de Salut Mar, Health and Experimental Sciences Department (CEXS), Universitat Pompeu Fabra (UPF), Barcelona Biomedical Research Park (PRBB), 08003 Barcelona, Spain; laura.manas@upf.edu (L.M.-G.); ac.penedo94@gmail.com (A.P.-V.); alopez2@imim.es (A.L.-P.); jorieke.deschrevel@kuleuven.be (J.D.); 2Centro de Investigación en Red de Enfermedades Respiratorias (CIBERES), Instituto de Salud Carlos III (ISCIII), 08003 Barcelona, Spain; 3Laboratory of Respiratory diseases and Thoracic Surgery, Department Chrometa, Catholic University of Leuven, B-3000 Leuven, Belgium; 4Scientific and Technical Department, Hospital del Mar-IMIM, 08003 Barcelona, Spain; xduran@imim.es

**Keywords:** muscle unloading model, cancer cachexia model, muscle damage, muscle proteolysis and autophagy, muscle regeneration

## Abstract

We hypothesized that in mice with lung cancer (LC)-induced cachexia, periods of immobilization of the hindlimb (7 and 15 days) may further aggravate the process of muscle mass loss and function. Mice were divided into seven groups (*n* = 10/group): (1) non-immobilized control mice, (2) 7-day unloaded mice (7-day I), (3) 15-day unloaded mice (15-day I), (4) 21-day LC-cachexia group (LC 21-days), (5) 30-day LC-cachexia group (LC 30-days), (6) 21-day LC-cachexia group besides 7 days of unloading (LC 21-days + 7-day I), (7) 30-day LC-cachexia group besides 15 days of unloading (LC 30-days + 15-day I). Physiological parameters, body weight, muscle and tumor weights, phenotype and morphometry, muscle damage (including troponin I), proteolytic and autophagy markers, and muscle regeneration markers were identified in gastrocnemius muscle. In LC-induced cachexia mice exposed to hindlimb unloading, gastrocnemius weight, limb strength, fast-twitch myofiber cross-sectional area, and muscle regeneration markers significantly decreased, while tumor weight and area, muscle damage (troponin), and proteolytic and autophagy markers increased. In gastrocnemius of cancer-cachectic mice exposed to unloading, severe muscle atrophy and impaired function was observed along with increased muscle proteolysis and autophagy, muscle damage, and impaired muscle regeneration.

## 1. Introduction

Skeletal muscle wasting is associated with chronic diseases including cancer. Cancer-induced cachexia is an invalidating condition that negatively influences the patients’ prognosis regardless of the status of the underlying tumor [1,2,3,4,5,6]. Muscle mass loss also takes place during periods of immobilization such as prolonged bed rest, microgravity, denervation, and critical illness [7,8,9,10,11]. The loss of muscle mass impairs muscle contractile function and atrophy mainly characterized by muscle weakness. Poor muscle performance entails an impairment in the patients’ daily life activities with a negative impact on their quality of life [8,12,13,14,15,16]. Furthermore, muscle mass loss due to cancer may be further aggravated by periods of muscle disuse. The specific contribution of each condition to the final muscle loss remains to be fully elucidated.

Several pathophysiological mechanisms such as oxidative stress, systemic inflammation, increased proteolysis, autophagy and apoptosis, and signaling pathways were shown to be involved in the process of muscle mass loss in limb muscles of cachectic patients with chronic conditions [17,18,19,20] and in animal models [18,19,21,22]. On the other hand, mechanisms related to muscle protein synthesis and regeneration may also be altered in cancer-induced cachexia and muscle wasting. 

Our group and others have demonstrated that in the gastrocnemius of mice exposed to different time-points of unilateral hindlimb immobilization, markers of muscle proteolysis and injury were increased [23,24,25]. Moreover, a decline in the number of progenitor muscle cells along with an upregulation of markers of activated satellite cells during a 7-day period of unloading of the limb muscles in mice was also demonstrated [26,27]. Whether the prolonged periods of immobilization may impair to a greater extent the muscle mass loss and dysfunction elicited by cancer-induced cachexia needs to be further investigated. Investigation of to what extent each particular condition—either cancer cachexia or prolonged immobilization—contributes to the wasting process of muscle mass also remains to be clarified. 

On the basis of this, we hypothesized that in mice with lung cancer (LC)-induced cachexia, periods of immobilization of the hindlimb (7 and 15 days) may further aggravate the process of muscle mass loss and function. Thus, the study objectives were that in mice with LC cachexia exposed to unloading for either 7 days or 15 days, the following parameters were analyzed: (1) body and gastrocnemius weights and limb strength, (2) muscle fiber type composition and morphometry and damage including troponin levels in blood, (3) markers of proteolysis and autophagy, and (4) identification of satellite cell subtypes and muscle regeneration markers. The experimental models used in the present investigation have been well-validated previously [18,19,21,28].

## 2. Results

### 2.1. Physiological Characteristics of the Study Animals 

#### 2.1.1. Non-Immobilized Controls versus Either Unloading or LC-Cachexia Conditions

Compared to non-immobilized (NI)-control animals, in either group of unloaded (7-day I and 15-day I) or in LC-cachexia (LC 21-days and LC 30-days) mice, total body and unloaded gastrocnemius weights and limb strength gain were significantly reduced (Table 1). No significant differences were observed in the weights of tibialis anterior, non-immobilized gastrocnemius, diaphragm, spleen (only in unloaded groups), lungs, or heart among the experimental groups (Table 1). However, diaphragm weight significantly decreased in LC 30-days animals compared to NI-control mice, while spleen weight was significantly greater in the two LC-cachexia groups compared to NI-control animals (Table 1). 

#### 2.1.2. LC-Cachexia+Unloading versus Unloading Conditions

Compared to unloaded (7-day I and 15-day I) mice, in animals with the two conditions (LC-cachexia and unloading), total body weight did not significantly differ between them, while gastrocnemius weight significantly decreased only in the LC 21-days + 7-day I group (−16% change), and limb strength gain significantly declined in both groups of mice (LC 21-days + 7-day I, −169% change and in LC 30-days + 15-day I, −167% change, respectively, Table 1). Non-immobilized muscle weights (gastrocnemius and tibialis anterior) significantly reduced (only in LC 30-days + 15-day I animals, −28% change and −17% change, respectively), while spleen weight significantly increased (+300% change and 503% change, in LC 21-days + 7-day I and LC 30-days + 15-day I, respectively, Table 1). Unloaded tibialis anterior, diaphragm, lungs, and heart weights did not significantly differ between animals bearing the two conditions (LC-cachexia and unloading) compared to unloaded mice (Table 1).

#### 2.1.3. LC-Cachexia+Unloading versus LC-Cachexia

Compared to LC-cachexia rodents, in mice with the two conditions (LC-cachexia and unloading), total body weight did not significantly differ between them, while unloaded gastrocnemius weight was significantly reduced only in LC 21-days + 7-day I (−22% change) and limb strength gain significantly decreased in LC 21-days + 7-day I and LC 30-days + 15-day I (−385% and −56% change, respectively, Table 1) mice. Tumor weight and tumor area significantly increased in LC 30-days + 15-day I compared to LC 30-days mice (+55% change, Table 1 and Figure 1). The weights of unloaded tibialis anterior, loaded muscles (gastrocnemius and tibialis anterior), diaphragm, spleen, lungs, and heart did not significantly differ among the experimental groups (Table 1).

### 2.2. Structural Phenotypic Characteristics

#### 2.2.1. Non-Immobilized Controls versus Either Unloading or LC-Cachexia Conditions

In gastrocnemius muscle, cross-sectional areas (CSA) of both slow- and fast-twitch muscle fibers were smaller in 15-day I mice and in both 7-day I and LC 30-days (fast-twitch myofibers) compared to the NI-controls (Table 2, Figure 2A,B). Proportions of muscle fiber types did not significantly differ among the study groups (Table 2, Figure 2A,B). Muscle structural abnormalities, internal nuclei, inflammatory cell counts, and other items significantly increased in both unloaded (7-day I and 15-day I cohorts) and LC-cachexia animals compared to NI-control mice (Table 2, Figure 2A,B).

#### 2.2.2. LC-Cachexia+Unloading versus Unloading Conditions 

Compared to unloaded animals, in the gastrocnemius of LC-cachexia and unloaded mice, CSA of fast-twitch fibers significantly decreased (LC 21-days + 7-day I, −13% change and in LC 30-days + 15-day I, −10% change), while CSA of slow-twitch fibers or fiber type proportions did not significantly differ (Table 2, Figure 2A,B). In the LC 21-days + 7-day I group, proportions of total muscle structural abnormalities, internal nuclei, and inflammatory cell counts significantly increased compared to unloaded mice (+34%, +80%, +248% change, respectively), while in the LC 30-days + 15-day I group, only internal nuclei counts were significantly increased (+103% change, Table 2, Figure 2A,B).

#### 2.2.3. LC-Cachexia+Unloading versus LC-Cachexia

Compared to LC-cachexia mice, in the gastrocnemius of LC-cachexia and unloaded animals, CSA of fast-twitch fibers were significantly reduced (LC 21-days + 7-day I, −14% change and in LC 30-days + 15-day I, −26% change), whereas CSA of slow-twitch fibers and fiber type proportions did not significantly differ (Table 2 and Figure 2B). Proportions of total muscle structural abnormalities, internal nuclei (only in LC 30-days + 15-day I), and inflammatory cell counts (only in LC 21-days + 7-day I) were significantly increased in LC-cachexia and unloaded animals compared to LC-cachexia mice (+30%, +41%, +55%, +45% change, respectively, Table 2 and Figure 2B).

### 2.3. Muscle Proteolysis and Proteolytic Markers

#### 2.3.1. Non-Immobilized Controls versus Either Unloading or LC-Cachexia Conditions 

Compared to NI-control animals, in unloaded and LC-cachexia mice, plasma troponin I levels significantly increased (Figure 3). Levels of MuRF-1 and proteasome were significantly greater in the 15-day I unloaded and LC 30-days animals than in NI-control mice (Figure 4A–C). No significant differences were seen in mice of the 7-day I and LC 21-days groups compared to NI-control mice for any of these markers (Figure 4A–C).

#### 2.3.2. LC-Cachexia+Unloading versus Unloading Conditions

Compared to unloaded (7-day I and 15-day I) mice, in gastrocnemius of animals with the two conditions (LC-cachexia and unloading), plasma troponin I levels were significantly higher in LC 30-days + 15-day I animals than in 15-day I mice (+67% change, Figure 3). Levels of MuRF-1 and proteasome did not significantly differ between LC-cachexia+unloaded animals compared to non-cachectic unloaded mice (Figure 4A–C).

#### 2.3.3. LC-Cachexia+Unloading versus LC-Cachexia

Compared to LC-cachexia animals, in mice with the two conditions (LC-cachexia and unloading), plasma troponin I levels were significantly increased in LC 30-days + 15-day I mice compared to LC 30-days rodents (+36% change, Figure 3). MuRF-1 and proteasome levels were significantly greater (only in LC 21-days + 7-day I, +72% change and +30% and +25% change in LC 21-days + 7-day I and LC 30-days + 15-day I, respectively for each marker) in both groups of LC-cachexia+unloaded rodents compared to both groups of LC-cachexia animals (Figure 4A–C).

### 2.4. Autophagy Markers

#### 2.4.1. Non-Immobilized Controls versus Either Unloading or LC-Cachexia Conditions

Compared to NI-control mice, in muscles of the two groups of unloaded mice and in LC 21-days rodents, beclin-1, p62, LC3-II/I, and cleaved caspase-3 protein levels did not significantly differ (Figure 5A–E). Moreover, beclin-1, LC3-II/I, and cleaved caspase-3 protein levels were significantly higher in muscles of the LC 30-days mice than in NI-controls, while no differences were detected in p62 protein levels between them (Figure 5A–E). 

#### 2.4.2. LC-Cachexia+Unloading versus Unloading Conditions

Compared to unloaded (7-day I and 15-day I) animals, in mice with the two conditions (LC-cachexia and unloading), beclin-1, p62, LC3-II/I, and cleaved caspase-3 protein levels did not significantly differ, while LC3-II/I levels (only in LC 30-days + 15-day I) significantly increased (+80% change, Figure 5A–E). 

#### 2.4.3. LC-Cachexia+Unloading versus LC-Cachexia

In gastrocnemius of mice with the two conditions (LC-cachexia and unloading), levels of the markers beclin-1, p62, LC3-II/I protein, and cleaved caspase-3 did not significantly differ among the study groups (Figure 5A–E). 

### 2.5. Satellite Cell Subtypes

#### 2.5.1. Non-Immobilized Controls versus Either Unloading or LC-Cachexia

Conditions Compared to NI-control animals, in muscles of the two unloaded groups of mice and in LC 30-days mice, activated (Pax-7+ and Myf-5+) satellite cell counts increased, while quiescent/regenerative potential (Pax-7+ and Myf-5-) cell numbers decreased, and total numbers of satellite cells did not significantly differ (Figure 6A,B and Figure 7A–C).

#### 2.5.2. LC-Cachexia+Unloading versus Unloading Conditions

Compared to unloaded animals, in muscles of mice with the two conditions (LC-cachexia and unloading), a significant decrease in the number of activated (Pax-7+ and Myf-5+) satellite cells (only in LC 30-days + 15-day I, −32% change) was observed, while counts of quiescent/regenerative potential (Pax-7+ and Myf-5-) cells and of total satellite cells did not significantly differ (Figure 6A,B and Figure 7A–C).

#### 2.5.3. LC-Cachexia+Unloading versus LC-Cachexia 

Compared to LC-cachexia mice, in muscles of LC-cachexia and unloaded rodents, numbers of activated (Pax-7+ and Myf-5+) satellite cells (only in LC 30-days + 15-day I) were significantly reduced (−29% change), while no significant modifications in quiescent/regenerative potential (Pax-7+ and Myf-5-) cells or total satellite cell numbers were detected (Figure 6B and Figure 7A–C).

### 2.6. Profile of Muscle Regeneration Markers

#### 2.6.1. Non-Immobilized Controls versus Either Unloading or LC-Cachexia Conditions

Compared to NI-control animals, in muscles of the two unloaded groups of animals and in the two LC-cachexia mice, myoblast (Pax-7+ and MyoD+) counts did not significantly differ, whereas the numbers of myocytes (MyoD+ and myogenin+) significantly decreased (Figure 8A–D and Figure 9A,B). Moreover, MyoD protein levels were significantly higher, while those of myogenin significantly declined in 7-day I mice compared to NI-control rodents (Figure 10A–C). No differences were detected between either 15-day I or LC 21-days, LC 30-days, and the NI-control mice for any of the MyoD or myogenin markers (Figure 10A–C).

#### 2.6.2. LC-Cachexia+Unloading versus Unloading Conditions

Compared to unloaded mice, in muscles of both groups of animals with the two conditions (LC-cachexia and unloading), the numbers of myoblasts (Pax-7+ and MyoD+) significantly decreased (−13% and −49% change in LC 21-days + 7-day I and LC 30-days + 15-day I, respectively), while those of myocytes (MyoD+ and myogenin+) did not significantly differ (Figure 8A–D and Figure 9A,B). Levels of the markers MyoD and myogenin did not vary among groups (Figure 10A–C).

#### 2.6.3. LC-Cachexia+Unloading versus LC-Cachexia

Compared to LC-cachexia mice, in muscles of LC-cachexia and unloaded rodents, a significant decline (only in the LC 30-days + 15-day I group) in the counts of myoblast (Pax-7+ and MyoD+, −49% change) was detected, whereas no significant differences in myocyte (MyoD+ and myogenin+) numbers were observed (Figure 8B,D and Figure 9A,B). Protein levels of MyoD and myogenin were not modified among the experimental groups of animals (Figure 10A–C).

## 3. Discussion

In the current investigation, the main findings were that in LC-induced cachexia mice exposed to different periods of unloading of the hindlimb muscles, gastrocnemius weight, limb strength gain, CSA of fast-twitch myofibers, and counts of activated satellite cells and myoblasts were significantly reduced, while tumor weight and area, muscle damage including systemic troponin levels, and proteolytic and autophagy markers increased in muscles. Furthermore, in gastrocnemius of mice exposed to either LC-induced cachexia or unloading compared to non-immobilized controls, total body and muscle weight, limb strength, CSA of fast-twitch muscle fibers, counts of quiescent satellite cells and myocytes, and myogenin levels significantly declined, whereas muscle damage including troponin I levels, proteolytic and autophagy (LC-cachexia) markers, activated satellite cell numbers, and MyoD levels were significantly greater. Collectively, these findings reveal that unloading induces a further increase in tumor burden, muscle damage, muscle atrophy—especially of the fast-twitch fibers—and muscle proteolysis in cancer cachectic mice. 

In hindlimb muscles of animals exposed to different periods of unloading, significant atrophy was also demonstrated in the target muscles and the proteasome system was shown to be a major proteolytic pathway in those experimental models [23,24,25]. In the current investigation, similar findings to those previously reported [23,24,25] have also been demonstrated in the gastrocnemius muscles of the animals exposed to 7 and 15 days of hindlimb unloading. Furthermore, a rise in the numbers of activated satellite cells and myoblasts was also detected in the gastrocnemius of the unloaded mice at both time-points (7-day I and 15-day I). These are similar findings to those reported in previous investigations [26,27]. Interestingly, a decline in the number of quiescent satellite cells was seen in the limb muscles of the same groups of mice. Similarly, these findings were also demonstrated in limb muscles of mice in investigations in which models of disuse muscle atrophy were used [26,27]. In addition, the number of myocytes was also significantly reduced in the unloaded gastrocnemius of the two time-points of immobilization. Likewise, similar findings were also reported in other studies [27]. Collectively, these are novel experimental approaches and findings that are discussed below.

In models of experimental cancer-associated cachexia and in patients with cachexia associated with chronic diseases including cancer, increased muscle wasting and loss of muscle function have been clearly demonstrated by our group and others [19,21,29,30]. The ubiquitin-proteasome system and autophagy and muscle damage have been consistently shown to vastly mediate the process of muscle protein loss in cancer cachectic patients and in animal models [19,21,22,30,31]. In addition, several drugs targeted to specific signaling pathways and metabolism were also shown to attenuate these mechanisms, while preventing additional muscle wasting in the animals [32,33,34,35]. In keeping with this, in the present study, in the gastrocnemius of the cachectic mice, similar findings were observed, such as increased expression of proteolytic markers, muscle damage, and loss of gastrocnemius muscle mass and strength along with muscle atrophy. Thus, very consistent results have been obtained in this study, in which the underlying biological mechanisms account for the alterations seen in the pathophysiologic outcomes: body and muscle weights, and strength.

Moreover, in the limb muscles of the cancer cachectic mice (LC-30-days), counts of activated satellite cells and myoblasts increased, whereas a decrease in the numbers of quiescent satellite cells was observed in the same muscles. A decline in the numbers of myocytes was also observed in the gastrocnemius of the cancer cachectic mice of the two time-points. These are similar findings to those previously shown [36,37]. Indeed, persistent expression of the self-renewal factor Pax7 was shown in in vivo and in vitro models of cancer cachexia in previous investigations [36,37]. 

These findings suggest that the regenerative potential (Pax7+/Myf-5 cells) was probably hampered in the cachectic muscles of the mice as this step is key for muscle repair and regeneration. Indeed, skeletal muscle regeneration potential was shown to be disrupted in cachectic C26 tumor cell-bearing mice [38]. Importantly, the authors concluded that cancer cachexia disturbed the skeletal muscle regenerative potential as a result of its ability to inhibit the muscle regeneration process [38].

Interestingly, tumor weight and area were significantly greater in the cachectic mice exposed to hindlimb unloading for 15 days. In cachectic mice exposed to 7 days, a similar trend was seen but no statistically significant difference was observed in these parameters, probably because the time-frame was short. These findings imply that immobilization favors tumor growth and the underlying mechanisms will have to be elucidated in future investigations specifically designed for that purpose. In fact, previous results have also suggested that reduced activity/deconditioning renders patients more prone to developing solid tumors [39,40,41]. Collectively, these are novel approaches and findings that suggest that the process of muscle fiber loss and wasting is potentiated when the two experimental conditions, cancer-induced cachexia and disuse muscle atrophy, take place simultaneously within the same muscles. In a similar fashion, expression of autophagy and signaling markers of muscle protein metabolism was also greater in limb muscles of cancer cachectic mice exposed to 14 days of denervation of the sciatic nerve [42]. Moreover, in patients with underlying muscle abnormalities including inactivity, cancer-associated cachexia was also more prominent [43]. Further studies will have to disentangle the biological mechanisms whereby disuse muscle atrophy enhances the loss of muscle mass and function in cancer cachexia models, both in patients and animals. As far as we are concerned, the current investigation addresses a novel relevant question that has been answered using novel experimental approaches in mice. Besides, the experimental conditions carried out herein reproduce to a great extent clinical scenarios of cancer-induced cachectic patients who may be exposed to prolonged periods of bed rest to receive different therapies (surgery, chemotherapy, etc.).

A significantly greater decline in gastrocnemius weight and limb strength was also observed in the cachectic mice exposed to hindlimb unloading than in non-immobilized cachectic mice. In addition, atrophy of fast-twitch myofibers along with higher levels of muscle damage including systemic troponin levels were also seen in cancer cachectic mice exposed to hindlimb unloading (especially the 15-day cohort). Moreover, protein levels of markers of proteolysis and autophagy were also significantly larger in the cancer cachectic mice exposed to unloading, even in animals belonging to the shorter cohort. 

In the current investigation, activated satellite cell counts were greater in the limb muscles of cancer cachexia mice as well as in those exposed to unloading compared to non-immobilized controls. However, in the gastrocnemius of mice bearing the two conditions, especially in LC 30-days+15-day I, a significant decline in the number of activated satellite cells was observed. Previous investigations also showed a rise in the number of muscle precursor cells in atrophying muscles, which was confirmed using electron microscopy [26,27], and in models of cancer cachexia [37,44,45]. Nonetheless, the numbers of activated satellite cells and myoblasts (Pax-7+/MyoD+ cells) were substantially reduced in the muscles of the cachectic animals exposed to 15 days of unloading, while the counts of quiescent satellite cells were similar to those detected in the gastrocnemius of either cancer cachexia or unloading conditions, separately. Furthermore, the number of myocytes (MyoD+/myogenin+ cells) was also reduced in the gastrocnemius of mice exposed to either unloading or cachexia conditions and in animals bearing the two conditions simultaneously. Taken together, these findings suggest that muscle regeneration is clearly disrupted when the two conditions take place concomitantly within the same muscle. This deleterious scenario prevents muscles from fully regenerating, a process which is absolutely required for muscle repair. This may also partly account for the severe muscle atrophy and functional implications of the hindlimb muscles, observed in the mice exposed to the two experimental conditions. The underlying mechanisms involved in the decrease in satellite cells in cachectic muscles during unloading conditions will have to be explored in future investigations. 

### Study Limitations

A limitation in the study is related to the measurement of grip strength using the four limbs in mice in which only one limb was immobilized. Nonetheless, this has probably had no impact on the study results as all the measurements were identically performed in all the animals used for comparison purposes. Furthermore, these procedures have been well-validated in previous investigations of our group [24,25,27,46]. Another potential limitation is related to the methodologies employed to assess muscle morphometry, in which representative fibers (a minimum of 100) were randomly measured in each muscle for all the experimental groups [21,24,46]. Other methodologies in which all fibers were counted from each muscle could be also be implemented in future investigations [47]. Nevertheless, as all the samples were analyzed identically for all the experimental groups, we do not believe that this may have had an impact on the study results. Another limitation might be related to the control groups of mice that could be used in the study. It is likely that an additional group of LC 15-days might have helped to clarify whether the results obtained in the LC 30-days + 15-day I could be solely attributable to the combined effects of LC-induced cachexia with disuse or to the larger size of tumor burden. Nonetheless, as the tumor is not visible up until day 15 or later, we considered thereafter that the ideal control groups for the LC 30-days + 15-day I cohort were the LC 21-days or the LC 30-days groups of mice in this specific investigation. Whether the analyses of other limb muscles may have yielded similar results could be of interest but will have to be investigated in future studies. The use of other models of lung cancer cachexia might also be considered in future investigations in the group, in which the results obtained in the present study would be further corroborated. 

## 4. Materials and Methods 

### 4.1. Animal Experiments

#### 4.1.1. Cells

The LP07 murine cell line was derived from a P07 lung tumor developed spontaneously in the lung of a BALB/c mouse and has been extensively characterized [48,49]. LP07 cells were maintained at 37 °C in a humidified, 5% CO2–air atmosphere in minimal essential media (MEM, Nuaille, France) supplemented with 10% fetal bovine serum (FBS, Bioser, GBO, Buenos Aires, Argentina), Penicillin/Streptomycin solution (Thermo Fisher Scientific Inc., Waltham, MA, USA) containing 10,000 U/mL of Penicillin, 10,000 µg/mL of Streptomycin, and 25 µg/mL of Fungizone (Amphotericin B, Thermo Fisher Scientific Inc.). To reach the number of cells necessary to be able to inject them into mice, we expanded the cells between 2 and 3 weeks. Subcultures were performed using trypsin-ethylenediaminetetraacetic acid (EDTA) 1× in phosphate-buffered saline (PBS, Thermo Fisher Scientific Inc.).

#### 4.1.2. Animals

In this study, BALB/c mice (10 weeks old, weight ~20 g, females) were purchased from Harlan Interfauna Ibérica SL (Barcelona, Spain). All the animals were maintained under a pathogen-free environment in the animal house facility at Barcelona Biomedical Research Park (PRBB), with a 12:12 h light:dark cycle. 

The study protocol is illustrated in Figure 11. An animal model with LC was developed through the subcutaneous inoculation of LP07 viable cells (4·105) resuspended in 0.2 mL of MEM in the left flank on the day 0. Moreover, unilateral hindlimb unloading was applied to specific groups of rodents, as previously reported, with the aim to mimic disuse muscle atrophy [24,25,27,46]. Essentially, clippers were used to shave the left hindlimb, which was subsequently protected with surgical tape. Microcentrifuge tubes of 1.5 mL (0.6 g) were used in the study. The cover and bottom lids were removed for the hindlimb to be introduced. The feet of the mice were kept in a plantar-flexed position in order to elicite the greatest degree of muscle atrophy [24,25,27,46]. The mice were able to move freely in the cages, even those wearing the plastic splints. The current experimental model has already been well validated, as shown in previous investigations [23,24,25]. For the study protocol, unloading was applied from day 15 (when the tumors are visible) up until day 21 (to conclude the 7 days of unloading) or until day 30 (to conclude the 15 days of unloading, Figure 11). Limb muscles of mice with the splint remained totally immobilized up to the end of the study protocol. For ethical reasons, we were not allowed to extend the study protocol beyond 30 days. 

The following groups of mice were investigated (*n* = 10/group, Figure 1): (1) non-immobilized (NI) control mice, without any period of unloading or tumor development, (NI-Control, 0.2 mL of MEM inoculated in mice left flank at day 0); (2) 7-day unloaded mice (7-day I, left hindlimb unloaded for seven consecutive days, 0.2 mL of MEM inoculated in mice left flank at day 0) starting the unloaded period at day 15; (3) 15-day unloaded mice (15-day I, left hindlimb unloaded for 15 consecutive days, 0.2 mL of MEM inoculated in mice left flank at day 0) starting the unloaded period at day 15; (4) 21-day lung cancer cachexia group (LC 21-days, inoculation of LP07 cells resuspended in 0.2 mL of MEM inoculated in mice left flank at day 0); (5) 30-day lung cancer cachexia group (LC 30-days, inoculation of LP07 cells resuspended in 0.2 mL of MEM inoculated in mice left flank at day 0); (6) 21-day lung cancer cachexia group besides 7 days of unloading (LC 21-days + 7-day I, left hindlimb unloaded for seven consecutive days, inoculation of LP07 cells resuspended in 0.2 mL of MEM inoculated in mice left flank at day 0) starting the unloaded period at day 15; (7) 30-day lung cancer cachexia group besides 15 days of unloading (LC 30-days + 15-day I, left hindlimb unloaded for seven consecutive days, inoculation of LP07 cells resuspended in 0.2 mL of MEM inoculated in mice left flank at day 0) starting the unloaded period at day 15. As the tumor was not visible up until day 15 or even later, we thereafter considered that the ideal control groups for the LC 30-days + 15-day I cohort were the LC 21-days or the LC 30-days groups of mice in this specific investigation. 

### 4.2. Ethics

Experiments involving the use of animals were all carried out in the animal facilities at our center PRBB. A controlled investigation was designed following the ethical regulations on animal experimentation in Europe (European Community Directive 2010/63/EU), Spain (Spanish Legislation, Real Decreto 53/2013, BOE 34/11370–11421), and the European Convention for the Protection of Vertebrate Animals Used for Experimental and Other Scientific Purposes (1986). The Animal Research Committee at PRBB approved the animal studies (Animal Welfare Department in Catalonia, Spain, EBP-17-0005, 2017/03/20).

### 4.3. Studies in Mice: In Vivo Measurements

The parameters of body weight and food intake were obtained daily for all the mice. Food and water were administered ad libitum. The parameter of limb strength was determined using a grip strength meter (Bioseb, Vitrolles Cedex, France) at the following time-points: day 0, at the end of the 21 days, and at the end of the 30 days as previously reported [18,21,24,25]. In this study, the four limbs were used to measure grip strength in all study groups. Limb strength gain was estimated as follows: (grip strength at the end of the study period-grip strength on day 0)/ grip strength on day 0 × 100 [18,21,24,25]. Tumor area was also measured daily using a specific caliper in all the animals, as previously reported [19,28]. Figure 1 represents the average area (mm^2^ units) of the tumors in each experimental group from day 15 (time at which they were initially visualized) up to day 21 or 30 (depending on the experimental group, time at which mice were sacrificed) every single day.

### 4.4. Mouse Sacrifice and Collection of the Samples

Following the periods of unloading and tumor development (21 or 30 days), all the animals were sacrificed in the animal facilities. Sodium pentobarbital (60 mg/kg, 0.1 mL) was administered five minutes prior to starting the sacrifice experiments. The pedal and blink reflexes were checked to confirm complete anesthesia. The gastrocnemius, tibialis anterior, and diaphragm muscles were dissected and collected from all the mice during sacrifice. Additionally, the heart, lungs, and spleen were dissected from all the experimental groups of mice during sacrifice. Liquid nitrogen was used to snap-freeze the muscle specimens, which were further stored in the −80 °C freezers. The entire muscles were obtained and used for the biological analyses. The gastrocnemius was divided transversally into two identical halves using a ruler. The two transversal halves were visually equal as to their muscle type composition. Furthermore, care was taken at the time of cutting the gastrocnemius in each animal for all the study groups in order to make sure that the same area of the muscle was cut for the further experiments. The halves obtained from each animal were monitored in size and fiber phenotype for all the study experiments in this investigation. One of the halves was frozen to conduct molecular experiments, while the other half was snap-frozen to be embedded in optimum cutting temperature (OCT) to assess muscle damage, fiber-typing, and morphometrical analyses.

### 4.5. Tissue Embedding

Muscle samples were fixed in 4% paraformaldehyde solution, pH 6.9 (EMD Millipore corporation, Billerica, MA, USA). They were subsequently embedded in increasing concentrations of sucrose. Additionally, the samples were also embedded in tissue-tek OCT compound (Sakura Finetek, Torrance, CA, USA). Thereafter, they were snap-frozen in 2-methyl-butane immersed in liquid nitrogen, as previously reported [50]. A cryostat-microtome (Leica CM3050S, Leica Biosystems, Wetzlar, Germany) was used to cut 10-µm frozen sections (−20 °C), to be thereafter mounted on glass slides.

### 4.6. Muscle Biology

#### 4.6.1. Muscle Fiber Typing and Morphometrical Analyses

Immunofluorescence procedures with specific antibodies (anti-myosin heavy chain (MyHC) I and anti-MyHC II, respectively) were used to determine slow- and fast-twitch muscle fibers in each preparation. The following protocol was used: muscle cross-sections were air-dried for thirty minutes and were rinsed with phosphate-buffered saline (PBS) for another fifteen minutes. Following rising, the sections were stored in cold methanol for six more minutes. Immediately afterwards, a 20-min period of boiling in a pressure cooker (10 mM citrate buffer (pH 6.0)) buffer followed. Samples were then cooled down to room temperature for two hours. The sections were then incubated with mouse IgG blocking reagent (MOM, Vector Laboratories, Burlingame, CA, USA) for one hour, as well as in blocking solution (3% Bovine serum albumin (BSA), 10% goat serum and 0.5% triton in PBS) for another hour. Later, they were incubated overnight with the mouse monoclonal anti-MyHC I antibody (ab11083, Abcam, Cambridge, UK), and anti-MyHC II antibody (ab51263, Abcam) prepared in blocking solution at 4 °C. Following incubation with the primary antibody and after rising with PBS, the sections were incubated with the corresponding secondary antibody and with 4′, 6-diamino-2-fenilindol (DAPI, which specifically stained deoxyribonucleic acid (DNA), allowing identification of all nuclei) for one hour at room temperature; anti-mouse IgG1 Alexa Fluor 488 antibody was also prepared in the blocking solution. Additionally, negative control experiments were carried out by omission of the primary antibody, and incubation of the muscle samples only with secondary antibody was also performed to confirm the specificity of each antibody.

Finally, the sections were mounted using 70% glycerol in 30% PBS. In two consecutive muscle cross-sections, muscle fibers positively stained with the anti-MyHC type I antibody or anti-MyHC type II were those that appeared as fluorescein isothiocyanate (FITC)-stained in green. The following parameters were determined in each muscle cross-section: mean least diameter, CSA, and the proportions of type I and type II using a fluorescence microscope (× 20 objective, Nikon Eclipse Ni, Nikon, Tokyo, Japan) coupled with and image-digitizing camera (Zyla 4.2 sCMOS camera, Andor, Belfast, UK) and the Image J software (National Institute of Health, available at http://rsb.info.nih.gov/ij/). At least 100 fibers (minimum of 150 fibers in all muscle preparations) were measured and counted, individually, in each muscle cross-section of all the animals. Methodologies employed in these experiments have all been well-validated in previous investigations from our group [19,21,24,25,27,30,51,52,53,54]

#### 4.6.2. Muscle Structure Abnormalities

Abnormalities in muscle structure were evaluated on ten-μm gastrocnemius OCT-embedded sections as previously reported [21,22,30,55]. Shortly, muscle cross-sections were stained with hematoxylin-eosin. Afterwards, images were captured using a light microscope (×40 objective, Olympus, Series BX50F3, Olympus Optical Co., Hamburg, Germany). To analyze the normal and abnormal muscle fraction, distinctive events were analyzed in images that were previously superimposed on a 63-square grid composed of a 7 × 9 rectangular pattern. The distinctive events to be analyzed on each square of the grid were categorized as follows: (0) no count, (1) normal muscle, (2) internal nucleus, (3) inflammatory cell, (4) lipofuscin, (5) abnormal viable fiber, (6) inflamed/necrotic fiber, and (7) blood vessel. The abnormal muscle fraction was calculated as a percentage of all the points that fell between the categories 2 and 6 relative to the total number of points overlaid counted on all viable fields. Internal nuclei were defined as the percentage of points that fell into category 2 relative to the total number of points counted. Inflammatory cells were identified as round cells with a nucleus clearly stained with hematoxylin, localized outside of the basal lamina of the muscle fibers. Inflammatory cells were computed as the percentage of points that fell into category 3 relative to the total number of points counted. Finally, a new category under the name of “other items” was established. This category grouped cells identified within the categories 4-6. Categories 0 and 7 were not considered in the counting. 

#### 4.6.3. Skeletal Muscle Troponin-I Levels in Plasma

In all the study groups of animals, skeletal muscle troponin-I levels were quantified in plasma samples, NI-Control, 7-day I, 15-day I, LC 21-days, LC 30-days, LC 21-days + 7-day I, LC 30-days + 15-day I, using a specific sandwich ELISA kit (Life Diagnostics Inc., West Chester, PA, USA) as previously described [24,46,56,57,58], at the end of the study protocol (day 21 or 30). Initially, samples and reagents were equilibrated to room temperature. A standard curve was always run with each assay run. Standards (100 μL) were loaded and the protocol was followed as indicated by the manufacturer’s instructions. All reagents used in these experiments were part of the specific ELISA kit. For all the study samples, equal volumes (100 μL total volume) of diluted plasma (1:3 dilution) were always loaded in duplicate onto the pre-coated ELISA-plate wells. Samples were incubated with 100 μL Horseradish peroxidase (HRP)-secondary antibody on an orbital micro-plate shaker at 150 rpm and 25 °C for one hour. The wells were then washed six times with the wash solution and incubated with 100 μL tetramethylbenzidine (TMB) reagent on an orbital micro-plate shaker at 150 rpm and at 25 °C for 20 min. Finally, the enzyme reaction was stopped by adding 100 μL stop solution to the wells. A microplate reader was used to read the absorbance in each plasma sample at 450 nm (655 nm reference filter). Intra-assay coefficients of variation for the plasma skeletal muscle troponin-I levels ranged from 2% to 10%. As all the samples were analyzed on the same day, no inter-assay coefficients of variation could be calculated.

#### 4.6.4. Immunoblotting of 1D Electrophoresis

Immunoblotting was used to identify the target markers in the study. Previously published procedures by our group and others were followed [21,24,25]. Muscle specimens from the gastrocnemius were homogenized in a specific buffer as previously reported [24]: 50 mM 4-(2-hydroxyethyl)-1-piperazineethanesulfonic acid (HEPES), 150 mM NaCl, 100 mM NaF, 10 mM Na pyrophosphate, 5 mM EDTA, 0.5% Triton-X, 2 micrograms/mL leupeptin, 100 micrograms/mL phenylmethanesulfonyl fluoride (PMSF), 2 micrograms/mL aprotinin, and 10 micrograms/mL pepstatin A.

All the experiments were carried out at 4 °C. Bradford methodologies were used to determine protein concentration in triplicate for each sample. BSA was used as the standard (Bio-Rad protein reagent, Bio-Rad Inc., Hercules, CA, USA). Equal amounts of total protein (ranging from 5 to 30 micrograms, according to antigen and antibody) from crude muscle homogenates were loaded for all the gels, as well as identical sample volumes/lanes. Muscle specimens were always run together for the seven study groups and were kept in the same order in the gels of mini-cell boxes for the sake of comparison. Four fresh 10-well mini-gels were run in the mini-cell boxes for each of the antigens. The samples were always transferred and detected together simultaneously for all the antigens analyzed in the investigation. Experiments were confirmed at least twice for all the antigens analyzed in the investigation.

Muscle proteins were separated by electrophoresis, subsequently transferred to polyvinylidene difluoride (PVDF) membranes, and then blocked with BSA to be incubated with selective primary antibodies overnight. Protein levels of the following markers and pathways were analyzed in the study muscles: structural proteins, proteolytic system, and autophagy. The following specific primary antibodies were used: ubiquitin-ligase muscle ring finger (MuRF)-1 (anti-MuRF-1 antibody, Santa Cruz Biotechnology, CA, USA), 20S proteasome subunit C8 (anti-C8 antibody, Biomol, Plymouth Meeting, PA, USA), beclin-1 (anti-beclin-1 antibody, Santa Cruz), nucleoporin p-62 (anti-p62/SQSTM1 antibody, Sigma-Aldrich, St. Louis, MO, USA), light chain (LC)3B (anti-LC3B antibody, Cell Signaling, Boston, MA, USA), cleaved-caspase-3 (anti-cleaved-caspase-3 antibody, Abcam), myogenic differentiation (MyoD, anti-MyoD antibody, Santa Cruz Biotechnology), myogenin (anti-myogenin antibody, Santa Cruz Biotechnology), and glyceraldehyde-3-phosphate dehydrogenase (GAPDH, anti-GAPDH antibody, Santa Cruz Biotechnology). HRP-conjugated secondary antibodies and a chemiluminescence kit were used to detect the different antigens on the membranes. Samples from the different groups were always detected in the same picture under identical exposure times for each of the antigens. Omission of the primary antibody and incubation of the membranes only with secondary antibodies were used to confirm the specificity of the different antibodies.

Alliance Q9 Advanced (Uvitec Cambridge, England, UK) was used to scan and analyze protein levels on the PVDF membranes for all the markers, including caspase-3, whose band was measured as the antibody directly identified cleaved caspase-3 [59]. Optical densities of specific bands were quantified using the ImageJ software (National Institute of Health, available at http://rsb.info.nih.gov/ij/). For each of the study antigens, mean values of the different samples (lanes) were the final optical densities used in the statistical analyses. GAPDH was used to validate equal protein loading in all the immunoblots. Positive controls were used to assess the specific band in ubiquitin-ligase atrogin-1 (MAFBx 293T lysate, Santa Cruz Biotechnology), MuRF-1 (MuRF-1 293T lysate, Santa Cruz Biotechnology). The ratio of LC3-II to LC3-I was calculated and is represented in graphs.

#### 4.6.5. Stripping Methodologies

Primary and secondary antibodies were stripped of proteins following a 30-min wash with a specific stripping solution (25 mM glycine, pH 2.0 and 1% sodium dodecyl sulfate (SDS)). Immediately afterwards, two consecutive 10-min washes of phosphate-buffered saline with tween (PBST) were performed at room temperature. Subsequently, membranes were blocked with bovine BSA to be reincubated with primary and secondary antibodies of the target marker following the methodologies described above.

#### 4.6.6. Satellite Cell Identification Using Immunofluorescence Microscopy

Immunofluorescence staining was used to detect satellite cells in both quiescent and activated states, myoblasts and myocytes using specific antibodies (see below). Briefly, muscle cross-sections were air-dried for 30 min and were rinsed with PBS for another 15 min. PBS was used to rinse the sections among the different incubation steps. After rinsing, the sections were put in cold methanol for six more minutes. Then, the sections were boiled using a hot bath in 0.1 M citrate buffer (pH 6.0) for 12 min and were then blocked with 10% goat serum in PBS for two hours. Subsequently, sections were incubated with MOM for 30 min. Afterwards, they were incubated overnight with four antibodies: paired box protein (Pax)-7, mouse monoclonal anti-Pax-7 antibody (Developmental Studies Hybridoma Bank, Iowa, IA, USA), myogenic factor (Myf)-5, rabbit polyclonal anti-Myf-5 antibody (Aviva Systems Biology, San Diego, CA, USA), rabbit polyclonal anti-MyoD antibody (Santa Cruz Biotechnology, Santa Cruz, CA, USA), and mouse monoclonal anti-myogenin antibody (Santa Cruz Biotechnology) prepared in an antibody solution (1% goat serum dissolved in PBS, at 4 °C. Anti-Pax-7 antibody alone was used to detect quiescent satellite cells, while the mixture of anti-Pax-7 and anti-Myf-5 antibodies detected committed satellite cells. The addition of quiescent and committed satellite cells corresponded to the total number of satellite cells. Moreover, the mixture of anti-Pax-7 and anti-MyoD was employed to detect myoblasts, whereas a mixture containing anti-MyoD and anti-myogenin detected myocytes. Following incubation with the primary antibodies and after rinsing with PBS, the sections were incubated at room temperature with the corresponding secondary antibodies for one hour: Alexa Fluor^®^ 488 AffiniPure goat Anti-mouse IgG, Fcγ Subclass 1 Specific and Alexa Fluor^®^ plus 555 goat anti-rabbit IgG (H+L, Thermo Fisher Scientific, Waltham, USA) also prepared in an antibody solution. Finally, the sections were mounted using the fluorescent mounting medium DAPI G-Fluoromount medium (Southern Biotech, Birmingham, AL, USA), which specifically stained deoxyribonucleic acid (DNA, allowing identification of all nuclei) in the muscle sections. Additionally, negative control experiments were carried out by omission of the primary antibody, and incubation of the muscle samples only with secondary antibody was also performed to confirm the specificity of each antibody. A fluorescence microscope (×40 objective, Nikon Eclipse Ni, Nikon, Tokyo, Japan) coupled with a digitizing camera was used to identify and count the number of the satellite cells (10 fields) in each study sample. Results were expressed as Pax-7+/Myf-5- (quiescent) satellite cells, Pax-7+/Myf-5+ (activated) satellite cells, the addition of both (as total satellite cells), or Pax-7+/MyoD+ (myoblast) and MyoD+/myogenin+(myocytes) to the total number of counted myonuclei in the 10 fields. 

### 4.7. Statistical Analysis

Shapiro–Wilk test was used to test the normality of the study variables. The results are presented as mean values (standard deviations). Results of the variables of food intake and percentage of change in total body weight, limb strength, muscle structure, and muscle damage are represented in Table 1 and Table 2. The molecular variables are represented in the figures. For each group, sample mean of the variable atrogin-1 as well as its estimated residual variance of two-way ANOVA (VarError = 0.00034653) were used to calculate sample size in the study. A minimum number of five mice in each group (35 in total) was necessary to achieve a minimum power of 80% with an alpha error equal to 0.05. The software Stata/MP release 15 (StataCorp LLC, College Station, TX, USA) was used for sample size calculation. For each specific period of unloading (either 7-day I or 15-day I), two-way analysis of variance (ANOVA) was performed using STATA separately. The following effects were analyzed: unloaded/non-immobilized mice, lung cancer cachectic/non-cachectic mice, and the interaction between unloading and cancer cachexia conditions in the animals for all the study variables. Moreover, potential differences between two groups were also analyzed using contrast of marginal linear predictions: (1) comparisons between non-immobilized controls and either unloading condition (7-day I or 15-day I), (2) comparisons between non-immobilized controls and either cancer cachexia cohort of mice (LC 21-days or LC 30-days), and (3) comparisons between any of the unloaded or cancer cachexia cohorts of mice and those bearing the two conditions (LC 21-days + 7-day I or LC 30-days + 15-day I). *p* ≤ 0.05 was established as the level of significance. Comparisons of the tumor area parameter were made among the different study groups at two time-points: last experimental day (day 21 or day 30) and at the initial time-point (day 15).

## 5. Conclusions

In gastrocnemius of cancer cachectic mice exposed to unloading, severe muscle atrophy and a decline in muscle strength were observed along with a significant rise in muscle proteolysis and autophagy, muscle damage, and impaired muscle regeneration. These results have clinical implications as cancer cachectic patients are frequently exposed to prolonged periods of bed rest.

## Figures and Tables

**Figure 1 ijms-21-08167-f001:**
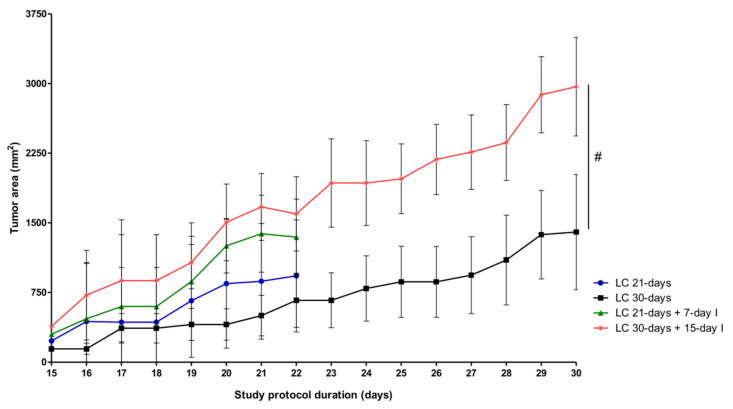
Mean values and standard deviation of subcutaneous tumor area (mm^2^) of the different groups of LC-cachexia mice and animals with both conditions (LC-cachexia and unloading) during the study protocol. Comparisons were made among the different study groups at two time-points: last experimental day (day 21 or day 30) and at the initial time-point (day 15). Definition of abbreviations: mm, millimeter; I, immobilization; LC, lung cancer. Statistical significance is represented as follows: ^#^
*p* < 0.05 between any group of LC-cachexia mice and their respective group of mice bearing the two conditions (LC-cachexia and unloading) at the end of the study period. No significant differences in tumor area were detected among the experimental groups.

**Figure 2 ijms-21-08167-f002:**
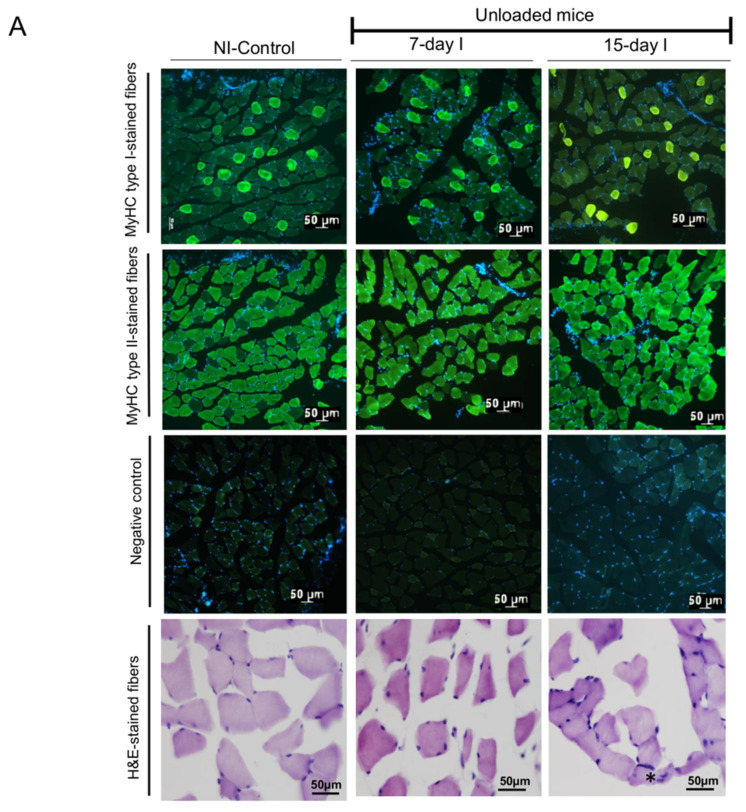
(**A**) Representative examples of the gastrocnemius muscle of the NI-control, 7-day I, and 15-day I groups of mice. (**B**) Representative examples of the gastrocnemius muscle of the LC 21-days, LC 30-days, LC 21-days + 7-day I, and LC 30-days + 15-day I groups of mice. Myofibers were stained in green; type I fibers (top panels), type II fibers (middle-up panels), and negative controls with no staining (middle-down panels). Hematoxylin-eosin stained muscle fibers are shown in the bottom panel. The arrowhead indicates an inflammatory cell and the asterisk points towards a fiber with an internal nucleus. Definition of abbreviations: MyHC myosin heavy chain; NI, non-immobilized; I, immobilization; H, hematoxylin; E, eoxin.

**Figure 3 ijms-21-08167-f003:**
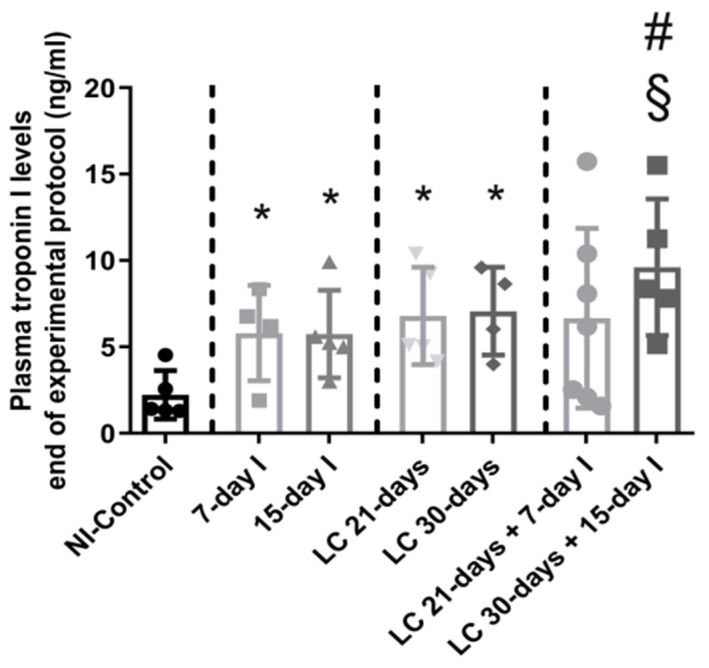
Mean values and standard deviation of the variable plasma troponin I (ng/mL) of the different study groups of mice. Definition of abbreviations: ng, nanogram; ml, milliliter; NI, non-immobilized; I, immobilization; LC, lung cancer. Statistical significance is represented as follows: * *p* < 0.05 between 7-day I, 15-day I, LC 21-days, LC 30-days animals, and the NI-control mice; # *p* < 0.05 between any group of LC-cachexia mice and their respective group of mice bearing the two conditions (LC-cachexia and unloading); ^§^
*p* < 0.05 between any group of unloaded mice and their respective group of mice bearing the two conditions (LC-cachexia and unloading).

**Figure 4 ijms-21-08167-f004:**
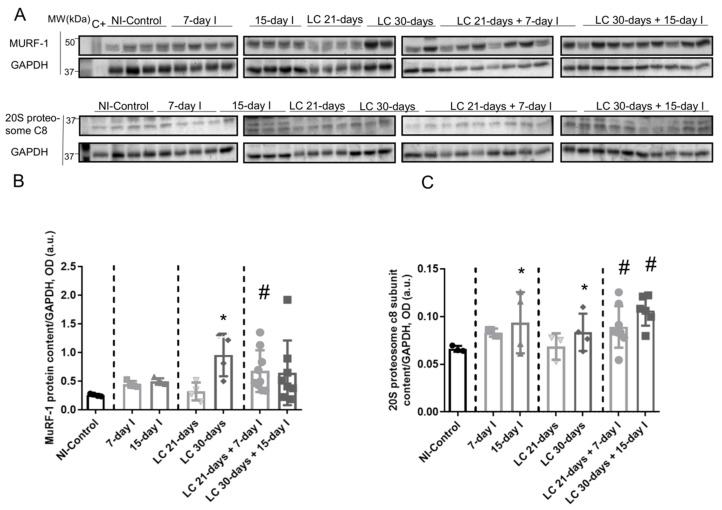
(**A**) Representative immunoblots of MuRF-1, 20S proteasome alpha subunit C8, and GAPDH proteins in the gastrocnemius muscle of all study groups of mice. (**B**) Mean values and standard deviation of MuRF-1 protein content in the gastrocnemius muscle of the different study groups of mice, as measured by optical densities in arbitrary units (OD, a.u.). Statistical significance is represented as follows: * *p* < 0.05 between 7-day I, 15-day I, LC 21-days, LC 30-days animals, and the NI-control mice; ^#^
*p* < 0.05 between any group of LC-cachexia mice and their respective group of mice bearing the two conditions (LC-cachexia and unloading). (**C**) Mean values and standard deviation of 20S proteasome c8 subunit protein content in the gastrocnemius muscle of the different study groups of mice, as measured by optical densities in arbitrary units (OD, a.u.). Statistical significance is represented as follows: * *p* < 0.05 between 7-day I, 15-day I, LC 21-days, LC 30-days animals, and the NI-control mice; ^#^
*p* < 0.05 between any group of LC-cachexia mice and their respective group of mice bearing the two conditions (LC-cachexia and unloading). Definition of abbreviations: MuRF-1, muscle RING-finger protein-1; GAPDH, glyceraldehyde-3-phosphate dehydrogenase; MW, molecular weight; kDa, kilodalton; C+, positive control; NI, non-immobilized; I, immobilization; LC, lung cancer.

**Figure 5 ijms-21-08167-f005:**
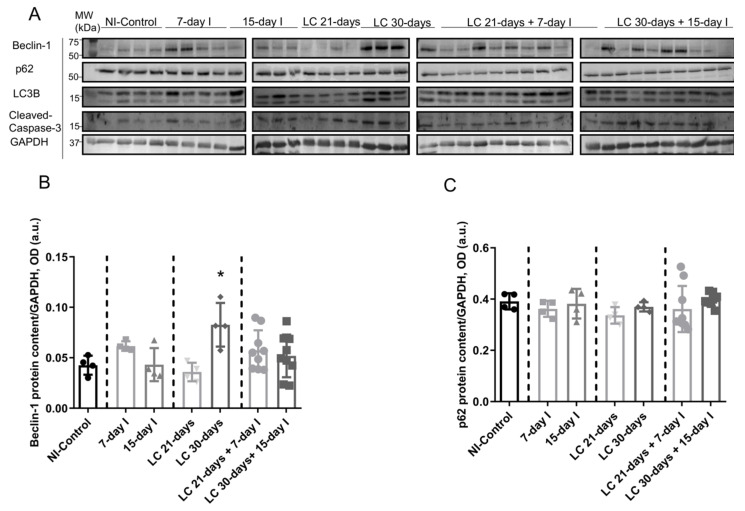
(**A**) Representative immunoblots of beclin-1, p62, LC3B, cleaved caspase-3, and GAPDH proteins in the gastrocnemius muscle of all study groups of mice. (**B**) Mean values and standard deviation of beclin-1 protein content in the gastrocnemius muscle of the different study groups of mice, as measured by optical densities in arbitrary units (OD, a.u.). Statistical significance is represented as follows: * *p* < 0.05 between 7-day I, 15-day I, LC 21-days, LC 30-days animals, and the NI-control mice. (**C**) Mean values and standard deviation of p62 protein content in the gastrocnemius muscle of the different study groups of mice, as measured by optical densities in arbitrary units (OD, a.u.). (**D**) Mean values and standard deviation of LC3B protein content in the gastrocnemius muscle of the different study groups of mice, as measured by optical densities in arbitrary units (OD, a.u.). Statistical significance is represented as follows: * *p* < 0.05 between 7-day I, 15-day I, LC 21-days, LC 30-days animals, and the NI-control mice; ^§^
*p* < 0.05 between any group of unloaded mice and their respective group of mice bearing the two conditions (LC-cachexia and unloading). (**E**) Mean values and standard deviation of cleaved caspase-3 protein content in the gastrocnemius muscle of the different study groups of mice, as measured by optical densities in arbitrary units (OD, a.u.). Statistical significance is represented as follows: * *p* < 0.05 between 7-day I, 15-day I, LC 21-days, LC 30-days animals, and the NI-control mice. Definition of abbreviations: p62, nucleoporin 62; LC3B, light chain 3 isoform B; GAPDH, glyceraldehyde-3-phosphate dehydrogenase; MW, molecular weight; kDa, kilodalton; NI, non-immobilized; I, immobilization; LC, lung cancer.

**Figure 6 ijms-21-08167-f006:**
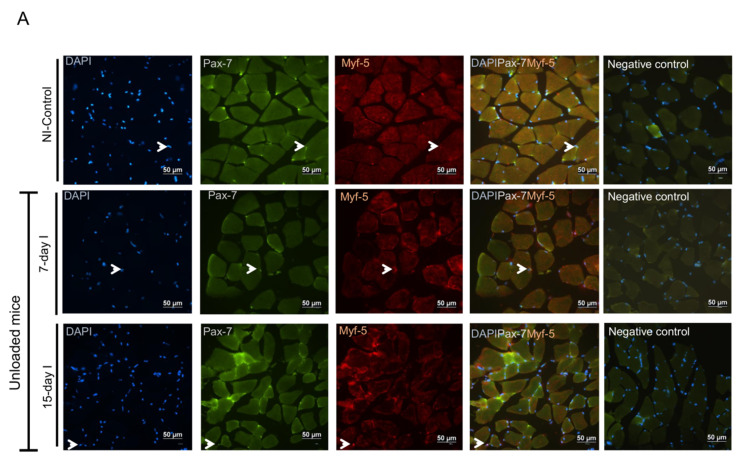
(**A**) Representative images of immunofluorescence staining of DAPI (left panels), Pax-7 (middle-left panels), Myf-5 (middle-right panels) and cells positively stained for both Pax-7 and Myf-5 markers (middle-right panels), and negative controls (right panels) in the gastrocnemius muscle of the NI-control mice and the unloaded mice (7-day I and 15-day I groups of animals). Arrows indicate double-stained nuclei for both Pax-7 and Myf-5 positive cells (activated satellite cells). (**B**) Representative images of immunofluorescence staining of DAPI (left panels), Pax-7 (middle-left panels), Myf-5 (middle-right panels) and cells positively stained for both Pax-7 and Myf-5 markers (middle-right panels), and negative control (right panels) in the gastrocnemius muscle of the LC-cachexia mice (LC 21-days, LC 30-days) and those bearing the two conditions (LC-cachexia and unloading, LC 21-days + 7-day I and LC 30-days + 15-day I groups of mice). Arrows indicate double-stained nuclei for both Pax-7 and Myf-5 positive cells (activated satellite cells). Definition of abbreviations: Pax-7, paired box -7; Myf-5, myogenic factor 5; DAPI, 4′,6-diamino-2-fenilindol; NI, non-immobilized; LC, lung cancer; I, immobilized.

**Figure 7 ijms-21-08167-f007:**
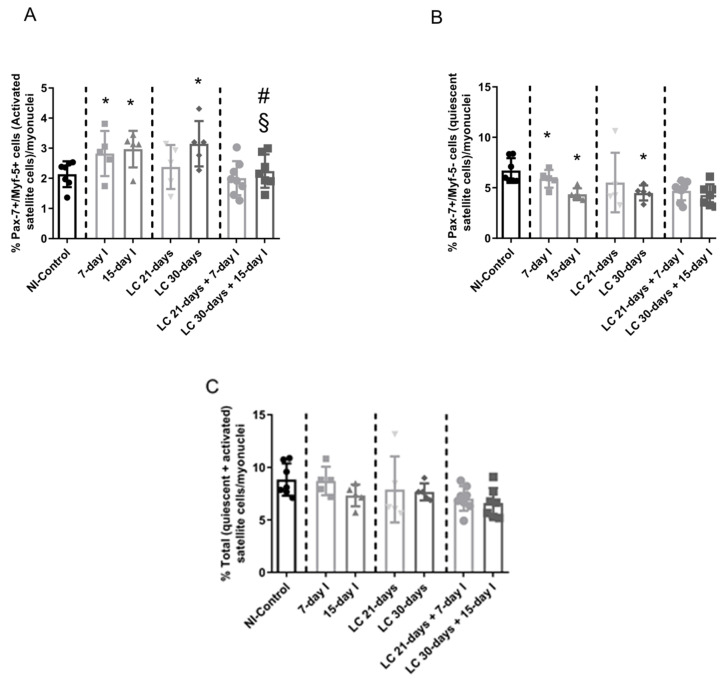
(**A**) Mean values and standard deviation of the percentage of activated satellite cell counts as identified by the number of Pax-7/Myf-5-positive cells in the gastrocnemius muscle of the different study groups of mice. Statistical significance is represented as follows: * *p* < 0.05 between 7-day I, 15-day I, LC 21-days, LC 30-days animals, and the NI-control mice; ^#^
*p* < 0.05 between any group of LC-cachexia mice and their respective group of mice bearing the two conditions (LC-cachexia and unloading); ^§^
*p* < 0.05 between any group of unloaded mice and their respective group of mice bearing the two conditions (LC-cachexia and unloading). (**B**) Mean values and standard deviation of the percentage of quiescent satellite cell counts as identified by the number of Pax7-positive cells (Myf-5-negative) in the gastrocnemius muscle of the different study groups of mice. Statistical significance is represented as follows: * *p* < 0.05 between 7-day I, 15-day I, LC 21-days, LC 30-days animals, and the NI-control mice. (**C**) Mean values and standard deviation of the percentage of total satellite cell counts as identified by the number of quiescent and activated satellite cells in the gastrocnemius muscle of the different study groups of mice. Statistical significance is represented as follows: * *p* < 0.05 between 7-day I, 15-day I, LC 21-days, LC 30-days animals, and the NI-control mice. Definition of abbreviations: Pax-7, paired box -7; Myf-5, myogenic factor 5; NI, non-immobilized; I, immobilization; LC, lung cancer.

**Figure 8 ijms-21-08167-f008:**
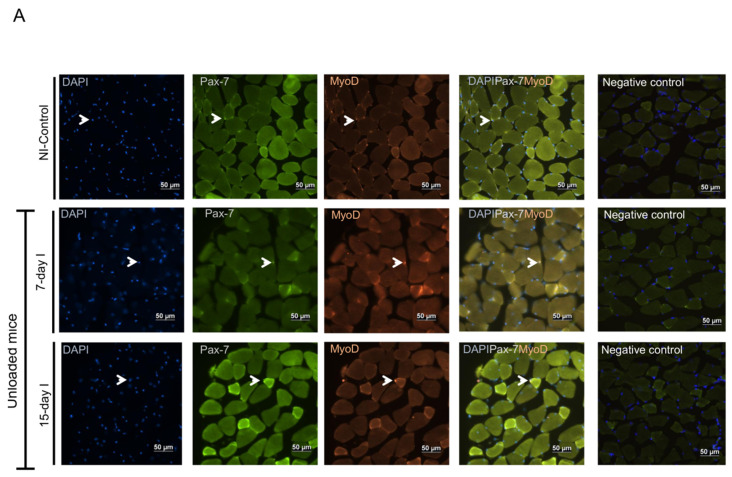
(**A**) Representative images of immunofluorescence staining of DAPI (left panels), Pax-7 (middle-left panels), MyoD (middle panels) and cells positively stained for both Pax-7 and MyoD markers (middle-right panels), and negative controls (right panels) in the gastrocnemius muscle of the NI-control mice and the unloaded mice (7-day I and 15-day I groups of animals). Arrows indicate double-stained nuclei for both Pax-7 and MyoD positive cells (myoblast). (**B**) Representative images of immunofluorescence staining of DAPI (left panels), Pax-7 (middle-left panels), MyoD (middle panels) and cells positively stained for both Pax-7 and MyoD markers (middle-right panels), and negative control (right panels) in the gastrocnemius muscle of the LC-cachexia mice (LC 21-days, LC 30-days) and those bearing the two conditions (LC-cachexia and unloading, LC 21-days + 7-day I and LC 30-days + 15-day I groups of mice). Arrows indicate double-stained nuclei for both Pax-7 and MyoD positive cells (myoblast). (**C**) Representative images of immunofluorescence staining of DAPI (left panels), myogenin (middle-left panels), MyoD (middle panels) and cells positively stained for both myogenin and MyoD markers (middle-right panels), and negative control (right panels) in the gastrocnemius muscle of the NI-control mice and the unloaded mice (7-day I and 15-day I groups of animals). Arrows indicate double-stained nuclei for both myogenin and MyoD positive cells (myocytes). (**D**) Representative images of immunofluorescence staining of DAPI (left panels), myogenin (middle-left panels), MyoD (middle panels) and cells positively stained for both myogenin and MyoD markers (middle-right panels), and negative control (right panels) in the gastrocnemius muscle of the LC-cachexia mice (LC 21-days, LC 30-days) and those bearing the two conditions (LC-cachexia and unloading, LC 21-days + 7-day I and LC 30-days + 15-day I groups of mice). Arrows indicate double-stained nuclei for both myogenin and MyoD positive cells (myocytes). *Definition of abbreviations*: Pax-7, paired box -7; MyoD, myogenic differentiation 1; DAPI, 4′,6-diamino-2-fenilindol; NI, non-immobilized; LC, lung cancer; I, immobilized.

**Figure 9 ijms-21-08167-f009:**
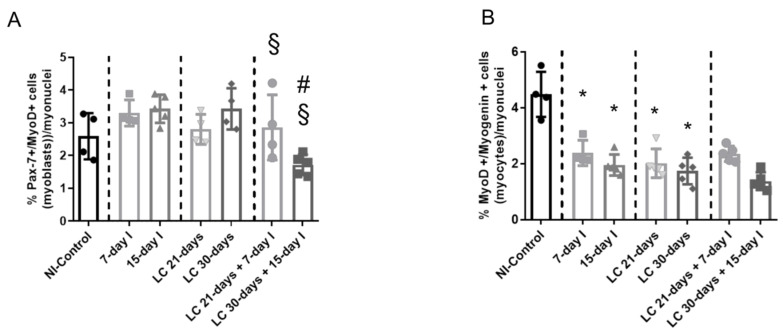
(**A**) Mean values and standard deviation of the percentage of myoblast counts as identified by the number of Pax-7/MyoD-positive cells in the gastrocnemius muscle of the different study groups of mice. Statistical significance is represented as follows: ^#^
*p* < 0.05 between any group of LC-cachexia mice and their respective group of mice bearing the two conditions (LC-cachexia and unloading); ^§^
*p* < 0.05 between any group of unloaded mice and their respective group of mice bearing the two conditions (LC-cachexia and unloading). (**B**) Mean values and standard deviation of the percentage of myocytes counts as identified by the number of MyoD/myogenin-positive cells in the gastrocnemius muscle of the different study groups of mice. Statistical significance is represented as follows: * *p* < 0.05 between 7-day I, 15-day I, LC 21-days, LC 30-days animals and the NI-control mice. Definition of abbreviations: Pax-7, paired box -7; MyoD, myogenic differentiation 1; NI, non-immobilized; I, immobilization; LC, lung cancer.

**Figure 10 ijms-21-08167-f010:**
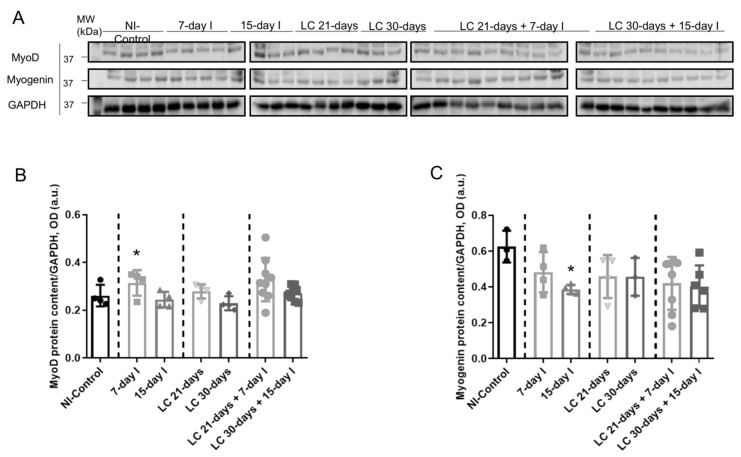
(**A**) Representative immunoblots of MyoD, myogenin, and GAPDH proteins in the gastrocnemius muscle of all study groups of mice. (**B**) Mean values and standard deviation of MyoD protein content in the gastrocnemius muscle of the different study groups of mice, as measured by optical densities in arbitrary units (OD, a.u.). Statistical significance is represented as follows: * *p* < 0.05 between 7-day I, 15-day I, LC 21-days, LC 30-days animals, and the NI-control mice; ^#^
*p* < 0.05 between any group of LC-cachexia mice and their respective group of mice bearing the two conditions (LC-cachexia and unloading); ^§^
*p* < 0.05 between any group of unloaded mice and their respective group of mice bearing the two conditions (LC-cachexia and unloading). **(C)** Mean values and standard deviation of myogenin protein content in the gastrocnemius muscle of the different study groups of mice, as measured by optical densities in arbitrary units (OD, a.u.). Statistical significance is represented as follows: * *p* < 0.05 between 7-day I, 15-day I, LC 21-days, LC 30-days animals, and the NI-control mice. Definition of abbreviations: MyoD, myogenic differentiation 1; GAPDH, glyceraldehyde-3-phosphate dehydrogenase; MW, molecular weight; kDa, kilodalton; NI, non-immobilized; I, immobilization; LC, lung cancer.

**Figure 11 ijms-21-08167-f011:**
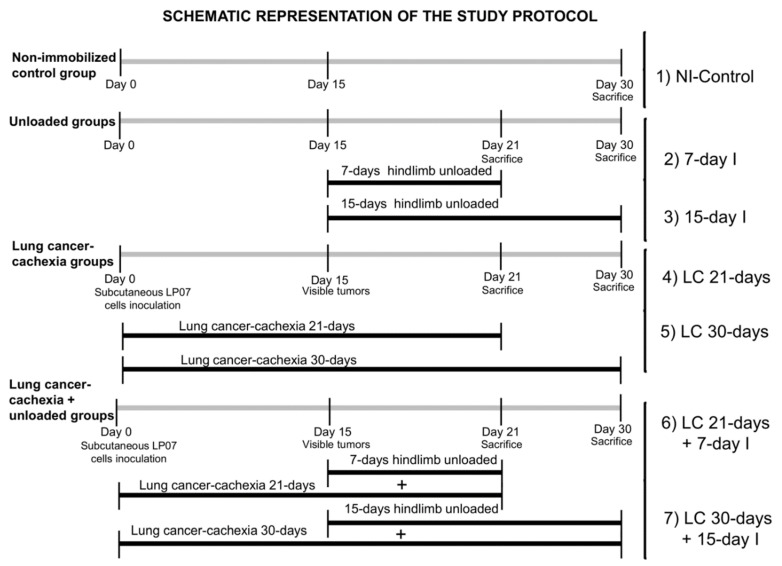
Graphical time-line representation of all the animal groups and the treatments administered to the mice in the study. Definition of abbreviations: NI, non-immobilized; I, immobilization; LC, lung cancer; MEM, minimum essential media.

**Table 1 ijms-21-08167-t001:** Physiological parameters in all the experimental groups of mice.

Physiological Parameters	NI-Control	7-Day I	15-Day I	LC 21-Days	LC 30-Days	LC 21-Days + 7-Day I	LC 30-Days + 15-Day I
**Total body weight gain (%)**	+7.38 (1.36)	−3.47 (1.90) *	−10.01 (1.74) **	−2.12 (1.93) *	−11.79 (2.83) **	−4.83 (1.87)	−14.53 (3.52)
***Unloaded limb muscle weights (g)***							
**Gastrocnemius**	0.125 (0.014)	0.106 (0.004) *	0.094 (0.002) **	0.115 (0.017) *	0.090 (0.017) **	0.089 (0.009) ^#,§§^	0.086 (0.016)
**Tibialis anterior**	0.045 (0.005)	0.042 (0.002)	0.040 (0.007)	0.044 (0.005)	0.042 (0.005)	0.041 (0.004)	0.038 (0.002)
***Contralateral-control limb muscle weights (g)***							
**Gastrocnemius**	0.112 (0.002)	0.108 (0.011)	0.110 (0.006)	0.110 (0.008)	0.085 (0.018)	0.095 (0.007)	0.079 (0.007) ^§^
**Tibialis anterior**	0.046 (0.009)	0.045 (0.008)	0.047 (0.005)	0.047 (0.005)	0.042 (0.013)	0.047 (0.008)	0.039 (0.007) ^§^
**Limb strenght gain (%)**	+7.24 (1.89)	−4.41 (1.95) **	−7.62 (1.80) **	−2.45 (1.56) *	−12.83 (3.21) **	−11.89 (3.84) ^#,§^	−20.04 (4.70) ^#,§^
**Tumor weight (g)**	NA	NA	NA	1.46 (0.68)	1.98 (0.10)	1.73 (0.59)	3.08 (1.13) ^#^
***Other organ weights (g)***							
**Diaphragm**	0.070 (0.006)	0.068 (0.008)	0.067 (0.011)	0.070 (0.004)	0.060 (0.010) *	0.064 (0.012)	0.058 (0.020)
**Spleen**	0.111 (0.007)	0.093 (0.010)	0.088 (0.001)	0.321 (0.090) *	0.531 (0.140) **	0.379 (0.090) ^§^	0.555 (0.130) ^§^
**Lungs**	0.167 (0.017)	0.162 (0.009)	0.158 (0.015)	0.186 (0.008)	0.220 (0.030)	0.173 (0.016)	0.225 (0.02)
**Heart**	0.115 (0.011)	0.107 (0.010)	0.108 (0.010)	0.100 (0.020)	0.106 (0.010)	0.108 (0.010)	0.106 (0.020)

Variables are presented as mean (standard deviation). Definition of abbreviations: NI, non-immobilized; LC, lung cancer; I, immobilization; g, grams; NA, not applicable. Statistical significance is represented as follows: * *p* < 0.05 and ** *p* < 0.01 between 7-day I, 15-day I, LC 21-days, LC 30-days animals, and the NI-control mice; ^#^
*p* < 0.05 between any group of LC-cachexia mice and their respective group of mice bearing the two conditions (LC-cachexia and unloading); ^§^
*p* < 0.05 and ^§§^
*p* < 0.01 between any group of unloaded mice and their respective group of mice bearing the two conditions (LC-cachexia and unloading).

**Table 2 ijms-21-08167-t002:** Structural characteristics of the gastrocnemius in all the experimental groups of animals.

Structural Characteristics	NI-Control	7-Day I	15-Day I	LC 21-Days	LC 30-Days	LC 21-Days + 7-Day I	LC 30-Days + 15-Day I
**Muscle fiber type, %**							
**Type I fibers**	12.50 (5.29)	15.91 (1.61)	14.04 (3.88)	14.87 (3.60)	17.01 (4.26)	19.04 (4.11)	15.35 (1.95)
**Type II fibers**	87.50 (5.29)	84.90 (1.69)	85.96 (3.88)	85.13 (3.60)	82.99 (4.26)	80.96 (4.11)	84.65 (1.95)
**Cross-sectional area, µm^2^**							
**type I fibers**	975.10 (169.18)	820.99 (110.23)	775.865 (180.30) *	951.61 (163.37)	796.24 (189.99)	766.28 (169.85)	615.41 (183.06)
**type II fibers**	1144 (154.25)	939.38 (159.62) *	641.36 (133.99) **	947.98 (149.49)	775.30 (56.29) *	819.46 (185.67) ^#,§^	575.05 (113.72) ^#,§^
**Total muscle structural abnormalities**	1.44 (0.40)	3.5 (0.86) **	3.82 (3.46) **	5.39 (0.80) **	5.69 (0.75) **	7.04 (0.48) ^#,§^	8.06 (2.63) ^#^
**Internal nuclei, %**	0.65 (0.32)	1.28 (0.29) *	1.52 (0.36) *	2.03 (0.55) *	1.99 (0.27) *	2.31 (0.43) ^§^	3.09 (1.22) ^#,§^
**Inflammatory cells, %**	0.73 (0.26)	1.11 (0.18) *	1.15 (3.15) *	2.66 (0.41) *	3.07 (0.96) *	3.87 (0.65) ^##,§§^	4.00 (1.45)
**Other items, %**	0.06 (0.09)	1.11 (0.55) *	1.15 (0.48) *	0.70 (0.22) *	0.63 (0.48) *	0.86 (0.45)	0.97 (0.66)

Variables are presented as mean (standard deviation). Definition of abbreviations: NI, non-immobilized; LC, lung cancer; I, immobilization; CSA, cross-sectional area. Statistical significance is represented as follows: * *p* < 0.05 and ** *p* < 0.01 between 7-day I, 15-day I, LC 21-days, LC 30-days animals, and the NI-control mice; ^#^
*p* < 0.05 and ^##^
*p* < 0.01 between any group of LC-cachexia mice and their respective group of mice bearing the two conditions (LC-cachexia and unloading); ^§^
*p* < 0.05 and ^§§^
*p* < 0.01 between any group of unloaded mice and their respective group of mice bearing the two conditions (LC-cachexia and unloading).

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
