# Peer review of "Prolonged Immobilization Exacerbates the Loss of Muscle Mass and Function Induced by Cancer-Associated Cachexia through Enhanced Proteolysis in Mice"

_ijms, 2020, doi:10.3390/ijms21218167_

Round 1

Reviewer 1 Report

concerns adequately addressed

Reviewer 2 Report

The revisions satisfactorily address my concerns. 

This manuscript is a resubmission of an earlier submission. The following is a list of the peer review reports and author responses from that submission.

Round 1

Reviewer 1 Report

The present study aimed at establishing the respective role of tumor burden and muscle immobilization on muscle wasting in tumor-bearing hosts. As such, this study addresses an important and very relevant question for cancer patients whom may experience both of these phenomena. The authors report that immobilization during the last 7 days of a 21-day tumor burden period further reduced muscle mass, type II fiber size and grip strength, and lead to muscle pathological appearance and increased atrogin-1 and MuRF1 protein expression compared to non-immobilized mice exposed to the same duration of tumor burden. Immobilization during the last 15 days of a 30-day tumor burden period further led to reduced grip strength and type II fiber size, and increased tumor mass, plasmatic level of troponin I and atrogin-1 protein expression compared to non-immobilized mice exposed to the same duration of tumor burden.

Overall, the current manuscript would greatly benefit from improved language clarity and concision as well as from a better flow. Typos should also be looked for. In its current form, several sentences are difficult to properly understand. Overall, the experimental design is adequate, but some of the methodology needs clarifications (see below). Given the presented images and data, some of the result interpretations are questionable (details provided below).

Major comments

  • The title of this work needs to be changed as activity per se was not measured. The present study indeed investigates the additive effect of tumor burden and muscle immobilization and not tumor burden and inactivity.

  • Given that tumor-bearing mice immobilized show bigger tumors (or tend to for the group immobilized for 7 days), an additional group of tumor-bearing mice is requested to draw definitive conclusions regarding the additive effect of immobilization and tumor burden on muscle wasting. Currently, it is not possible to discard that the differences observed in the various outcomes between LC30-days and LC30-days + 15-days I are not simply resulting from the greater tumor burden induced by the bigger tumor size in the LC30-days + 15-days I group.

  • It is unclear why masses of other muscles were not reported. Indeed, the protocol used for the limb immobilization here would also have resulted in immobilizing all lower leg muscles in addition to the gastrocnemius. Masses of at least some of these additional muscles would be interesting to report. In addition, masses from other limb muscles that were not immobilized would also be of interest to report. Although some compensation mechanisms could have occurred, this effect should be detectable if it exists by comparing these muscle masses from non-immobilized tumor-bearing mice to immobilized tumor-bearing mice. Additional outcomes such as fiber size could also be informative. Heart, spleen and fat masses would also be informative to evaluate tumor-burden.

  • Can the authors clarify if the immobilized limb remained immobilized at the later time points where muscle strength was recorded? In addition, recording of the four-limb strength while only one limb was immobilized represent a limitation of the present study and this should be highlighted in the manuscript.

  • The authors state that a portion of the gastrocnemius muscle was used for histology. Could the authors provide details on which portion they chose (i.e. red vs. white) for which outcomes and how they made sure to consistently sample the same portion for all mice. This is really important given the heterogeneity of the gastrocnemius fiber composition.

  • To be accurate, fiber size and fiber typology analyses need to be done on much more than 100 fibers. In the gastrocnemius muscle, 100 fibers represent less than 5% of the total fibers. Desgeorges et al. 2019 (Skeletal Muscle, PMC6323738) indeed demonstrated the importance of analyzing all fibers of a given cross sections to avoid bias in the data (see their Fig. 3).

  • Figure 2:
    • The representative images do not always depict the averaged data shown in Table 2. For instance, it looks like type I fibers are, if anything, bigger in the 15-day I compared to the NI-control group. Similarly, it looks like there are more type II fibers in 7-day I compared to the NI-control group. Moreover, based on the H&E images, it is not obvious that the immobilization protocol did indeed induced atrophy.
    • Important differences in the cytosolic staining can be observed, including a very weak and pale stain in the NI-control group.
    • Based on several panels, it appears that some sections were not fully cross sectional (e.g., 15-day I H&E or LC21 days). At minimum, authros should present minimal ferret diameters instead of CSA to partly account for this.
    • It remains unclear how the authors could confidently identify non-muscle cell nuclei as nuclei from inflammatory cells without staining for specific markers. Also, based on the representative H&E image provided for the NI-control group, it looks like there are as many or more non-muscle cell nuclei in this group compared to any of the other groups.
    • For comparison purposes, it would be easier to have all panels for a given marker/outcome side by side for all groups.

  • Figure 4:
    • It does not appear that GAPDH represent a stable loading control across different groups here. Also, it seems that histograms in panels B and C represent raw expression rather than expression normalized to GAPDH. Could the authors present normalized values, either to GAPDH if it can be validated as a stable loading control and/or to total protein?
    • Were the statistical analyses performed on raw data or on normalized ones?
    • Based on both the cropped and uncropped blot, it is difficult to see which band was analyzed for each given sample for atrogin-1. Please provide a blot with each band analyzed indicated. It also looks like atrogin-1 antibody might not be highly specific, the positive control is indeed showing three bands and the different samples show multiple bands very close in term of molecular weight to each other. This piece of data does not yield strong confidence in the interpretation of the atrogin-1 level.
    • Although discussed in the text, the authors to not present quantification of the 20S proteasome C8 nor of the total ubiquitinated protein. These quantifications should be added although I am concerned that total ubiquitinated protein could not be quantified for all samples given the blots.
    • Is the highly expressed band shown on the uncropped 20S Proteasome C8 blot representing GAPDH? In link with this, how many times was a single membrane stripped and re-stained?
  • Figure 5:
    • As in Figure 4, GAPDH does not seem to represent a good loading control as differences across groups can be observed.
    • It is difficult to see how a clean quantification was done for cleaved-caspase 3 given the high background and presence of specks. In addition, it would be best practice to present cleaved-caspase 3 in relation to caspase 3.
    • Please provide normalized expression to a loading control.
  • Figure 6:
    • Based on the current provided images, I have concerns regarding the specificity of the Myf5 stain. I cannot see positive staining where indicated and believe that the overall apparent colocalization (yellow) on merged images is mainly the result of the Pax7 positive nuclei overlapping with non-specific red staining.
    • I also have slight concerns regarding Pax7 stain specificity. In LC21-days + 7-day I, we clearly see one nucleus being pointed by the arrow on the DAPI image, whereas two ‘nuclei’ can be observed on the Pax7 image.
    • Overall, in the actual state, I do not believe that those data are of good quality enough to draw any conclusions regarding satellite cell numbers and/or activation.

Minor comments

  • The objectives stated at the end of the introduction are not objectives but outcomes. This is one instance where language editing would be required.
  • Throughout the results, please present an outcome for all groups and then move on. The section on satellite cell subtypes is indeed following this and is the clearer section of the results.
  • Please present the data of Table 1 and 2 in figures and depict individual data points. This will allow the reader to grasp the data more easily, while being able to clearly see data distribution/variability. Table 1 data could easily be included into figure 1 with tumor masses for instance.
  • For all histograms, please depict all individual data. This is best practice and allow the reader to fully make his/her own informed interpretation.
  • In Figure 1, is the “#” representing tumor masses being different throughout the progression of the cancer or only at endpoint?
  • One of the authors’ interpretations of their finding is that “inactivity favors tumor growth” (l. 371). Yet, similar than for the title, this interpretation needs to be tuned down as the present study did not measure daily activity. According to the authors themselves, mice wearing the plastic splint were moving freely in their cage. It remains therefore to be established if their activity level was reduced or not compared to non-immobilized tumor-bearing mice.
  • Line 426: I believe the authors are referring to Figure 8.
  • Line 427: is “(4.105)” referring to the number of cells injected, i.e. 4E5? If not, please clarify and provide the number of cells injected.
  • Lines 438-442: I believe that this section should be moved to the end of the discussion
  • On figure 8, it would improve clarity if the name of the groups were better aligned to their corresponding timeline. This scheme is overall very dense to read. If possible, it would be good to simplify it, or at least space out the different groups. Also, please indicate that non-tumor-bearing mice were injected with MEM on day 0.
  • Line 464: write out PRBB
  • Line 471: Please indicate what are “every time-point”. Is it daily?
  • Histology methods: please clarify that some samples were fixed in 4% PFA and then embedded in sucrose, whereas some others where embedded in OCT and frozen. Currently, it looks like the same sample underwent PFA fixation, sucrose embedding and OCT embedding.
  • Muscle structure abnormalities: Although the authors present an interesting method to classify muscle as more or less pathological, could they provide references supporting the validity of this method. How were inflammatory cells identified? Please provide a better ‘name’ to describe the currently called “other items” outcome.
  • Number of internalized nuclei are generally presented either as number per field of view or number of internalized nuclei per total number of nuclei.

Author Response

Reviewer # 1:

C1

The present study aimed at establishing the respective role of tumor burden and muscle immobilization on muscle wasting in tumor-bearing hosts. As such, this study addresses an important and very relevant question for cancer patients whom may experience both of these phenomena. The authors report that immobilization during the last 7 days of a 21-day tumor burden period further reduced muscle mass, type II fiber size and grip strength, and lead to muscle pathological appearance and increased atrogin-1 and MuRF1 protein expression compared to non-immobilized mice exposed to the same duration of tumor burden. Immobilization during the last 15 days of a 30-day tumor burden period further led to reduced grip strength and type II fiber size, and increased tumor mass, plasmatic level of troponin I and atrogin-1 protein expression compared to non-immobilized mice exposed to the same duration of tumor burden.

R1

We are thankful to this reviewer for having reviewed our manuscript and for the insightful comments. We have revised the entire manuscript according to all the reviewers’ concerns. Moreover, we have provided appropriate responses to each of the reviewer’s concerns in this letter.

We are also grateful to the reviewer for considering that the study is very important and that addresses a relevant question for cancer patients whom experience both of these phenomena. We have answered all the reviewer’s concerns below.

C2

Overall, the current manuscript would greatly benefit from improved language clarity and concision as well as from a better flow. Typos should also be looked for. In its current form, several sentences are difficult to properly understand. Overall, the experimental design is adequate, but some of the methodology needs clarifications (see below). Given the presented images and data, some of the result interpretations are questionable (details provided below).

R2

We thank for these comments. We are grateful to the reviewer for considering that the experimental design is adequate and for requesting an improvement in the methodology. Moreover, we have corrected the language of the entire manuscript. We have used the track changes tool of the Word software.

Major comments

C3

  • The title of this work needs to be changed as activity per se was not measured. The present study indeed investigates the additive effect of tumor burden and muscle immobilization and not tumor burden and inactivity.

R3

We thank the reviewer for this comment. We have modified the title: the word “inactivity” has been replaced with immobilization (See title page in the revised manuscript).

C3 

  • Given that tumor-bearing mice immobilized show bigger tumors (or tend to for the group immobilized for 7 days), an additional group of tumor-bearing mice is requested to draw definitive conclusions regarding the additive effect of immobilization and tumor burden on muscle wasting. Currently, it is not possible to discard that the differences observed in the various outcomes between LC30-days and LC30-days + 15-days I are not simply resulting from the greater tumor burden induced by the bigger tumor size in the LC30-days + 15-days I group.

R3

We thank the reviewer for this comment. In the study, two additional groups of tumor-bearing mice were also included as can be seen in all the Figures (LC-21 days and LC-30 days). These were the control groups of the experimental groups: LC-21 days + 7 day I and LC-30 days + 15 day I, respectively. Moreover, two more additional groups were also included in the study: 7-day I and 15-day I, respectively. On top of this, an additional group with no exposure to immobilization or no lung cancer was included for the sake of comparisons. In conclusion, we do believe that all the required/necessary experimental groups were, indeed, included in the present investigation. We do hope that the reviewer will also agree with this.

  • C4
  • It is unclear why masses of other muscles were not reported. Indeed, the protocol used for the limb immobilization here would also have resulted in immobilizing all lower leg muscles in addition to the gastrocnemius. Masses of at least some of these additional muscles would be interesting to report. In addition, masses from other limb muscles that were not immobilized would also be of interest to report. Although some compensation mechanisms could have occurred, this effect should be detectable if it exists by comparing these muscle masses from non-immobilized tumor-bearing mice to immobilized tumor-bearing mice. Additional outcomes such as fiber size could also be informative. Heart, spleen and fat masses would also be informative to evaluate tumor-burden.

R4

We thank the reviewer for these comments. In revised Table 1, information on the tibialis anterior and gastrocnemius are shown for the immobilized limb and the control non-immobilized limb. Additionally, the weights of: heart, lungs, spleen, and diaphragm have also been included in revised Table 1 as requested by the reviewer. In revised Results, this information has also been described  (See lines 109-114, 152-158, and 165-167).

  • C5
  • Can the authors clarify if the immobilized limb remained immobilized at the later time points where muscle strength was recorded?
  • R5
  • We thank the reviewer for this comment. The limbs with the splint remained totally immobilized up to the end of the study protocol right before measuring limb strength for all the animals (See lines 879-880). We understand the reviewer’s concerns regarding the use of the four limbs in the measurements of grip strength. However, as these procedures were identically applied to all the animals, we believe that this may have had no significant effects on the outcomes. These aspects have been discussed in the revised manuscript (See lines 769-772).

 C7

  • The authors state that a portion of the gastrocnemius muscle was used for histology. Could the authors provide details on which portion they chose (i.e. red vs. white) for which outcomes and how they made sure to consistently sample the same portion for all mice. This is really important given the heterogeneity of the gastrocnemius fiber composition.
  •  

R7

We thank the reviewer for these comments. As the muscle mass (average 100 mg, depending on the experimental groups) is relatively small, the entire gastrocnemius muscle was used for all the experimental procedures for all the study groups in the current investigation. Thus, no distinction was made between red and white fibers (See lines 938-941 in the revised manuscript). All the fibers were present in the homogenates and used in the histological preparations.

 C8

  • To be accurate, fiber size and fiber typology analyses need to be done on much more than 100 fibers. In the gastrocnemius muscle, 100 fibers represent less than 5% of the total fibers. Desgeorges et al. 2019 (Skeletal Muscle, PMC6323738) indeed demonstrated the importance of analyzing all fibers of a given cross sections to avoid bias in the data (see their Fig. 3).

R8

We thank the reviewer for this comment. While we acknowledge the quality of the study results and methodologies used in the study referred by the reviewer, the procedures used in the current investigation have been extremely well-validated in numerous investigations of our group and others (See lines 1003-1004 and references 19,21,24,25,27,30,51-54).

C9

  • Figure 2:
    • The representative images do not always depict the averaged data shown in Table 2. For instance, it looks like type I fibers are, if anything, bigger in the 15-day I compared to the NI-control group. Similarly, it looks like there are more type II fibers in 7-day I compared to the NI-control group. Moreover, based on the H&E images, it is not obvious that the immobilization protocol did indeed induced atrophy.
    • Important differences in the cytosolic staining can be observed, including a very weak and pale stain in the NI-control group.
    • Based on several panels, it appears that some sections were not fully cross sectional (e.g., 15-day I H&E or LC21 days). At minimum, authros should present minimal ferret diameters instead of CSA to partly account for this.
    • It remains unclear how the authors could confidently identify non-muscle cell nuclei as nuclei from inflammatory cells without staining for specific markers. Also, based on the representative H&E image provided for the NI-control group, it looks like there are as many or more non-muscle cell nuclei in this group compared to any of the other groups.
    • For comparison purposes, it would be easier to have all panels for a given marker/outcome side by side for all groups.

R9

We thank the reviewer for these comments. We have modified all the images corresponding to fiber typing and structure in order to improve the quality of the results and to match with the numbers reported in Table 2. Methodologies corresponding to all these concepts have also been improved in the revised manuscript (See lines 1020-1024). The methodologies employed in the present study have all been well-validated as demonstrated in previous publications by our group and others (See references 21,22,30,51).

Finally, images have been structured and organized in such a way that can be easily seen and assessed by any potential reader of the study, in the event of the manuscript being accepted for publication in the Journal.

C10

  • Figure 4:
    • It does not appear that GAPDH represent a stable loading control across different groups here. Also, it seems that histograms in panels B and C represent raw expression rather than expression normalized to GAPDH. Could the authors present normalized values, either to GAPDH if it can be validated as a stable loading control and/or to total protein?
    • Were the statistical analyses performed on raw data or on normalized ones?

C11

We thank the reviewer for these comments. Values represented in the ordinate axis were those corresponding to the ratio of the optical densities of each marker in each group to those of GAPDH. However, we acknowledge that this information was not explicit in the formerly submitted manuscript. Thus, caption in the ordinate axis has been modified for all the graphs depicted in the revised manuscript version (See Figures 4, 5, and 10). Statistical analyses of comparisons between groups as to GAPDH levels yielded P values all greater than 0.050 (P=0.485, P=0.93, P=0.96, P=1, etc.). As no statistically significant differences were found in GAPDH levels among the study groups, GAPDH was an appropriate loading control in this study. Finally, statistical analyses were all conducted on the variables corrected for GAPDH expression.

C12

  • Based on both the cropped and uncropped blot, it is difficult to see which band was analyzed for each given sample for atrogin-1. Please provide a blot with each band analyzed indicated. It also looks like atrogin-1 antibody might not be highly specific, the positive control is indeed showing three bands and the different samples show multiple bands very close in term of molecular weight to each other. This piece of data does not yield strong confidence in the interpretation of the atrogin-1 level.

R12

We thank the reviewer for these comments. In the uncropped blots a large blue arrow was included with the aim to indicate which band corresponded to each of the target antigens along with a positive control when available for each Figure. In the specific case of atrogin-1 the darkest band obtained in the positive control corresponded to a molecular weight of 42 kDa, which is actually the molecular weight of atrogin-1.

C13

  • Although discussed in the text, the authors to not present quantification of the 20S proteasome C8 nor of the total ubiquitinated protein. These quantifications should be added although I am concerned that total ubiquitinated protein could not be quantified for all samples given the blots.

R13

We thank the reviewer for this comment. We regret to say that Figure 4 (panels D & E) in the formerly and revised manuscript versions corresponded, indeed, to measurements of the optical densities and statistical analyses in each experimental group of the markers 20S proteasome and total protein ubiquitination levels. The blots were, indeed, quantified.

C14

  • Is the highly expressed band shown on the uncropped 20S Proteasome C8 blot representing GAPDH? In link with this, how many times was a single membrane stripped and re-stained?

R14

We thank the reviewer for this comment. We have not analyzed that band as this was not the target band in the experiment. Proteomics analyses should be conducted in order to specifically analyze/identify that band. As this was not the purpose in the study, we have not analyzed the band.

C15

  • Figure 5:
    • As in Figure 4, GAPDH does not seem to represent a good loading control as differences across groups can be observed.

R15

We thank the reviewer for these comments. Values represented in the ordinate axis were those corresponding to the ratio of the optical densities in each group to those of GAPDH. However, we acknowledge that this information was not explicit in the formerly submitted manuscript. Thus, caption in the ordinate axis has been modified for all the graphs depicted in the revised manuscript version (See Figures 4, 5, and 10). Statistical analyses of comparisons between groups as to GAPDH levels yielded P values all greater than 0.050 (P=0.485, P=0.93, P=0.96, P=1, etc.). As no statistically significant differences were found in GAPDH levels among the study groups, GAPDH was an appropriate loading control in this study.

C16

  • It is difficult to see how a clean quantification was done for cleaved-caspase 3 given the high background and presence of specks. In addition, it would be best practice to present cleaved-caspase 3 in relation to caspase 3.

R16

We thank the reviewer for this comment. The quantification was done following previous methodologies using the UVITEC system (See lines 1079 and 1089-1090, references 55 and 56). The antibody used to analyze caspase-3 detected, indeed, cleaved caspase-3 in the immunoblots. Thus, no ratio was calculated.

C17

  • Please provide normalized expression to a loading control.
  •  
  • R17
  • We thank the reviewer for this comment. We invite the reviewer to see images in Figures 4, 5, and 10 in the revised manuscript.

  • C18
  • Figure 6:
    • Based on the current provided images, I have concerns regarding the specificity of the Myf5 stain. I cannot see positive staining where indicated and believe that the overall apparent colocalization (yellow) on merged images is mainly the result of the Pax7 positive nuclei overlapping with non-specific red staining.
    • I also have slight concerns regarding Pax7 stain specificity. In LC21-days + 7-day I, we clearly see one nucleus being pointed by the arrow on the DAPI image, whereas two ‘nuclei’ can be observed on the Pax7 image.
    • Overall, in the actual state, I do not believe that those data are of good quality enough to draw any conclusions regarding satellite cell numbers and/or activation.

R18

We thank the reviewer for these comments. We have modified the images in order to improve the quality of the Figures. Moreover, in the revised manuscript, additional experiments have been carried out and results added in Figures 8-10 (immunofluorescence and immunoblotting experiments) to confirm data.

Minor comments

C19

  • The objectives stated at the end of the introduction are not objectives but outcomes. This is one instance where language editing would be required.

R19

We thank the reviewer for this comment. Outcomes were not precisely established in the Objectives section of the manuscript. The formulation of an outcome would have implied the definition of a specific increase/improvement/decrease/reduction of a given parameter following an intervention. As this was not the case, objectives were correctly defined in the manuscript (both formerly submitted and revised versions). Moreover, outcomes are usually defined in clinical studies as in RCTs, which is not the case of the present investigation.

  • C20
  • Throughout the results, please present an outcome for all groups and then move on. The section on satellite cell subtypes is indeed following this and is the clearer section of the results.

R20

We thank the reviewer for these comments. Identical structure of the paragraphs and description of the study results have been followed in all the Results section. A thorough Editing has been performed throughout the entire manuscript including the Results section (See 104-581 in the revised manuscript).

C21

  • Please present the data of Table 1 and 2 in figures and depict individual data points. This will allow the reader to grasp the data more easily, while being able to clearly see data distribution/variability. Table 1 data could easily be included into figure 1 with tumor masses for instance.

R21

We thank the reviewer for these comments. As the number of results included in Tables 1 & 2 is huge, many different figures would have been required in order to depict each of the analyzed parameters in individual figures of all the experimental groups.  Thus, Tables 1 & 2 have been kept the same in the revised version.

Individual data points have been included in each figure for all the markers (See Figures in the revised manuscript).

C22

  • For all histograms, please depict all individual data. This is best practice and allow the reader to fully make his/her own informed interpretation.

R22

We thank the reviewer for this comment. Individual data points have been included in each figure for all the markers (See Figures in the revised manuscript).

  • C23
  • In Figure 1, is the “#” representing tumor masses being different throughout the progression of the cancer or only at endpoint?

R23

We thank the reviewer for this comment. Tumor masses were not represented in Figure 1. In fact, Figure 1 represents the average area (mmm2 units) of the tumors in each experimental group from day 15 (time at which they were initially visualized) up to day 21 or 30 (depending on the experimental group, time at which mice were sacrificed) during every single day. Comparisons were made among the different study groups (See statistical analyses) at two time-points: last experimental day (day 21 or day 30, significance depicted in the graph) and at the initial time-point (day 15, no significant differences as expected). This information was present in the Methods section (formerly and revised manuscript versions, see lines 1165-1167) and legend to Figure 1.

C24

  • One of the authors’ interpretations of their finding is that “inactivity favors tumor growth” (l. 371). Yet, similar than for the title, this interpretation needs to be tuned down as the present study did not measure daily activity. According to the authors themselves, mice wearing the plastic splint were moving freely in their cage. It remains therefore to be established if their activity level was reduced or not compared to non-immobilized tumor-bearing mice.

R24

We thank the reviewer for this comment. The title has been modified in the revised manuscript. Moreover, the text has been entirely modified in order to avoid the use of the word “activity/inactivity” throughout the entire manuscript (See modified text). The use of immobilization and/or unloading has been preferred in the revised manuscript.

C25

  • Line 426: I believe the authors are referring to Figure 8.

R25

We thank the reviewer for this comment. Figure 11 has been indicated in the revised manuscript.

  • C26
  • Line 427: is “(4.105)” referring to the number of cells injected, i.e. 4E5? If not, please clarify and provide the number of cells injected.

R26

Yes indeed, this is the commonly used nomenclature to define numbers of cells.

C27

  • Lines 438-442: I believe that this section should be moved to the end of the discussion

R27

We thank the reviewer for this remark. However, this is not the authors’ responsibility as we did not include the Conclusions in the middle of the Methods. This must have been a mistake related to the formatting of the manuscript by the editorial office. In the revised, manuscript version, we have used the plain Word document (See lines 1168-1173 in the revised manuscript at the end of the Methods section, as requested by the Journal).

C28

  • On figure 8, it would improve clarity if the name of the groups were better aligned to their corresponding timeline. This scheme is overall very dense to read. If possible, it would be good to simplify it, or at least space out the different groups. Also, please indicate that non-tumor-bearing mice were injected with MEM on day 0.

R28

We thank the reviewer for this comment. Modifications have been made in Figure 11 in the revised manuscript.

  •  
  • C29
  • Line 464: write out PRBB

R29

PRBB definition was defined when first mentioned already (See line 866 in the revised manuscript).

C30

  • Line 471: Please indicate what are “every time-point”. Is it daily?

R30

We thank the reviewer for this comment. The expression has been changed to “daily”, which represents better what was actually done.

C31

  • Histology methods: please clarify that some samples were fixed in 4% PFA and then embedded in sucrose, whereas some others where embedded in OCT and frozen. Currently, it looks like the same sample underwent PFA fixation, sucrose embedding and OCT embedding.

R31

We thank the reviewer for this comment. The reviewer has understood very well the procedures employed to fix the target tissues for the histological analyses (See lines 944-951 in the revised manuscript).

C32

  • Muscle structure abnormalities: Although the authors present an interesting method to classify muscle as more or less pathological, could they provide references supporting the validity of this method. How were inflammatory cells identified? Please provide a better ‘name’ to describe the currently called “other items” outcome.

R32

We thank the reviewer for this comment. Additional information and references have been included in the Methods section (See lines 1020-1025 in the revised Methods). The original reference (MacGowan et al. AJRCCM 2001) has also been cited in the revised manuscript.

C33

  • Number of internalized nuclei are generally presented either as number per field of view or number of internalized nuclei per total number of nuclei.

R33

We thank the reviewer for this comment. Explanations on how internal nuclei were counted and expressed were given in the formerly submitted manuscript and in the revised text (See lines 1020-1025 in the revised Methods).

Reviewer 2 Report

This is worthy paper looks at the interaction of cancer cachexia and inactivity on muscle structure metabolism and function. It showed additive effects. The study was underpowered, so comparisons that were not significant remain uncertain. The discussion was slightly superficial and could be updated. Please comment on which aspects of pathways shown to be important in muscle wasting associated with the two conditions under study in this paper are operative and overlapping. as well as novel factors affecting the relationships studied.

Pleas reorder the manuscript in a more conventional format with methods followed by results before finishing with discussion and conclusions.

Author Response

Reviewer # 2:

C1

This is worthy paper looks at the interaction of cancer cachexia and inactivity on muscle structure metabolism and function. It showed additive effects. The study was underpowered, so comparisons that were not significant remain uncertain. The discussion was slightly superficial and could be updated. Please comment on which aspects of pathways shown to be important in muscle wasting associated with the two conditions under study in this paper are operative and overlapping. as well as novel factors affecting the relationships studied.

R1

We are thankful to this reviewer for having reviewed our manuscript and for the insightful comments. We have revised the entire manuscript according to all the reviewers’ concerns. Moreover, we have provided appropriate responses to each of the reviewer’s concerns in this letter.

We are also grateful to the reviewer for considering that the study is worthy.

The investigation was not underpowered as described in the Methods section (under the Statistical analyses subheading). A total of 5 mice/group was required to attain a minimum power of 80% with an alpha risk =0.05. As 10 animals were used in each experimental the study was not underpowered at all. Thus, non-significant comparisons should be interpreted as such.

The Discussion has been modified in order to make it more profound. Moreover, pathways common to the two conditions have been discussed right after overall summary of the main findings (See revised Discussion).

C2

Pleas reorder the manuscript in a more conventional format with methods followed by results before finishing with discussion and conclusions.

R2

We thank the reviewer for these comments. It should be noted that the order of the Discussion and Conclusions must remain the same as this is the order established by the Journal.

Reviewer 3 Report

The manuscript “Prolonged Inactivity Exacerbates the Loss of Muscle Mass and Function Induced by Cancer-Associated Cachexia Through Enhanced Proteolysis in Mice” seeks to explore the hypothesis that prolonged inactivity (hindlimb immobilization) worsens wasting phenotypes associated with cancer cachexia. This is an interesting area of study as many cancer patents become progressively less active as cachexia worsens and/or as a consequence of therapy, etc. Overall, the study is well designed with appropriate control and experimental groups and the authors provide a balanced analysis of cachectic phenotypes. Experimental power seems adequate. Specific comments/concerns are provided below:

  • The analyses are almost exclusively focused on GR muscle. Can the authors please justify why this muscle was selected and perhaps include data from other skeletal muscle types?
  • Subcutaneous models of cancer [cachexia] are becoming increasingly less popular due to newer models more accurately reflecting tumor and/or wasting biology. While it would be unreasonable to request repeating all of these experiments in a more physiologically relevant model, validation of a few key assays in an additional cancer cachexia model (ideally not subcutaneous but rather orthotopic or autochthonous) would greatly strengthen this study.
  • In general, the embedding/sectioning/imaging of muscle tissue is subpar. The H/E images in particular are fairly low quality. Higher quality sections/images are needed.
  • What are “other items” in table 2? Does this correspond to the aggregate of the non-inflammatory cell counts listed in the methods?
  • How were immune cells identified? The methods do not seem to specify.
  • Figures 3,4,5,7: Please show individual data points for transparency.
  • The satellite cell analysis is questionable. The background staining is excessively high and some arrows do not even seem to be pointing to bona-fide satellite cells (ie. Fig6, LC 30d sample). Given that the changes are, at best, pretty subtle, it might be necessary to show additional evidence of impaired myogenesis/regeneration (ie later-stage myoblast/myocyte marker staining, regeneration assays, etc) otherwise, the relevance of these changes is unclear.
  • The Myf5 staining is particularly messy; if the authors are having trouble with this stain, there are other ways to discriminate activated vs quiescent SCs using MyoD or Ki67 antibodies that work much better than Myf5 in tissue sections.
  • Numerous typos throughout main text and figure legends. Please fix.

Author Response

Reviewer # 3:

C1

The manuscript “Prolonged Inactivity Exacerbates the Loss of Muscle Mass and Function Induced by Cancer-Associated Cachexia Through Enhanced Proteolysis in Mice” seeks to explore the hypothesis that prolonged inactivity (hindlimb immobilization) worsens wasting phenotypes associated with cancer cachexia. This is an interesting area of study as many cancer patents become progressively less active as cachexia worsens and/or as a consequence of therapy, etc. Overall, the study is well designed with appropriate control and experimental groups and the authors provide a balanced analysis of cachectic phenotypes. Experimental power seems adequate.

R1

We are thankful to this reviewer for having reviewed our manuscript and for the insightful comments. We have revised the entire manuscript according to all the reviewers’ concerns. We are also very grateful to the reviewer for considering that the study is interesting, well-designed with appropriate control and experimental groups, well-balanced analysis of the cachectic phenotypes, and that experimental power was adequate. Moreover, estimations of the sample size have been described in the Methods section under the Statistical analysis subheading in the revised manuscript.

Besided, we have provided appropriate responses to each of the reviewer’s concerns in this letter.

Specific comments/concerns are provided below:

C2

  • The analyses are almost exclusively focused on GR muscle. Can the authors please justify why this muscle was selected and perhaps include data from other skeletal muscle types?

R2

We thank the reviewer for this comment. Gastrocnemius muscle was chosen in the study as this is a very representative limb muscle of a mixture of slow- and fast-twitch fibers in mice. Additionally, due to its relatively large size many different types of experiments can be conducted as it was the case herein. The analysis of muscles of smaller size would have been interesting but they will have to be analyzed in future studies. This has been discussed under the “Study limitations” subheading in the revised Discussion (See lines 779-780). Furthermore, the weights of tibialis anterior and diaphragm muscles have been listed in revised Table 1.

C3

  • Subcutaneous models of cancer [cachexia] are becoming increasingly less popular due to newer models more accurately reflecting tumor and/or wasting biology. While it would be unreasonable to request repeating all of these experiments in a more physiologically relevant model, validation of a few key assays in an additional cancer cachexia model (ideally not subcutaneous but rather orthotopic or autochthonous) would greatly strengthen this study.

R3

We thank the reviewer for these insightful comments. We also understand the concerns raised by the reviewer, with whom we also agree. Nonetheless, at this stage of the peer-review (we were initially granted 10 days and have finally invested 20 days in the revision of the manuscript) the performance of additional experiments would be simply impossible: ethical approval of a new project, set up of the animal experiments, performance of the animal experiments, biological experiments, collection of data, statistical analyses, data interpretation, and manuscript writing. All these steps would entail the use of several human resources and facilities for several months and even years that would preclude the potential publication of the current study, if deemed acceptable by the Editors and the reviewers.  We are very thankful, however, to the reviewer for these suggestions that are very welcome and may be implemented in future investigations of our group. See lines 780-782 under the “Study limitations” section in the revised Discussion.

  • C4
  • In general, the embedding/sectioning/imaging of muscle tissue is subpar. The H/E images in particular are fairly low quality. Higher quality sections/images are needed.

R4

We thank the reviewer for this comment. The pictures have been replaced with fresh images in the revised manuscript version (See images in Figure 2).

  • C5
  • What are “other items” in table 2? Does this correspond to the aggregate of the non-inflammatory cell counts listed in the methods?

R5

We thank the reviewer for this comment. Additional information has been included in the Methods section (See lines 1020-1025 in the revised Methods).

  • C6
  • How were immune cells identified? The methods do not seem to specify.

R6

We thank the reviewer for this comment. In this study, inflammatory cells were identified as round cells, with a nucleus clearly stained with hematoxylin, localized outside of the basal lamina of the muscle fibers. These details have been described in the revised Methods (See lines 1020-1022).

C7

  • Figures 3,4,5,7: Please show individual data points for transparency.

R7

We thank the reviewer for this comment. Figures have been modified for the sake of transparency (See Figures 3-10 in the revised manuscript).

C8

  • The satellite cell analysis is questionable. The background staining is excessively high and some arrows do not even seem to be pointing to bona-fide satellite cells (ie. Fig6, LC 30d sample). Given that the changes are, at best, pretty subtle, it might be necessary to show additional evidence of impaired myogenesis/regeneration (ie later-stage myoblast/myocyte marker staining, regeneration assays, etc) otherwise, the relevance of these changes is unclear.

R8

We thank the reviewer for these concerns. These types of experiments were not easy to handle, especially in this type of mouse background. Despite these concerns, an effort has been made to identify better images to be included in Figure 6 of the revised manuscript. Moreover, additional experiments (immunoblotting and immunofluorescence) have been conducted in order to further identify/confirm the expression of the markers: MyoD and myogenin (See Figures 8-10 in the revised manuscript).

C9

  • The Myf5 staining is particularly messy; if the authors are having trouble with this stain, there are other ways to discriminate activated vs quiescent SCs using MyoD or Ki67 antibodies that work much better than Myf5 in tissue sections.

R9

We thank the reviewer for this comment. As abovementioned expression levels of the markers MyoD and myogenin have been analyzed in the revised manuscript version (See Figures 8-10 in the revised manuscript).

C10

  • Numerous typos throughout main text and figure legends. Please fix.

R10

We thank very much the reviewer for this comment. We have corrected the entire manuscript text for typos and general editing of English language.

Round 2

Reviewer 1 Report

Although I appreciate the effort to improve the image quality, the quality of the images remains below the quality required to analyze and interpret those data, raising strong concerns regarding the results presented here. For example, in panel B of Figure 2, a high autofluorescence can be observed on several of the representative images. This is also the case for Figure 6 and 8, where it definitely precludes any follow up analyses of the stain. Furthermore, based on the representative images from Figure 2, it appears that different muscle areas were used to quantify MHCI and MHCII percentage. In addition, the H&E images suggest that several muscle sections were not cross-section given the shape of the fibers. I regret to say that these concerns with the images provided raise red flags regarding the data supporting most of the authors conclusions.

Additional comments

Previous comment: Given that tumor-bearing mice immobilized show bigger tumors (or tend to for the group immobilized for 7 days), an additional group of tumor-bearing mice is requested to draw definitive conclusions regarding the additive effect of immobilization and tumor burden on muscle wasting. Currently, it is not possible to discard that the differences observed in the various outcomes between LC30-days and LC30-days + 15-days I are not simply resulting from the greater tumor burden induced by the bigger tumor size in the LC30-days + 15-days I group.

Authors’ Response: We thank the reviewer for this comment. In the study, two additional groups of tumor-bearing mice were also included as can be seen in all the Figures (LC-21 days and LC-30 days). These were the control groups of the experimental groups: LC-21 days + 7 day I and LC-30 days + 15 day I, respectively. Moreover, two more additional groups were also included in the study: 7-day I and 15-day I, respectively. On top of this, an additional group with no exposure to immobilization or no lung cancer was included for the sake of comparisons. In conclusion, we do believe that all the required/necessary experimental groups were, indeed, included in the present investigation. We do hope that the reviewer will also agree with this.

New comment: Although I do appreciate that several control groups were included in this study, I do still believe that an additional control group would be required to make definitive conclusions on immobilization exacerbating muscle wasting. Based on the current data, it can be stated that immobilization associates with increased tumor mass when comparing the LC-30 days + 15-days I vs. LC-30 days groups. It is therefore not possible to discard that the differences observed between those two groups in several outcomes are simply caused by the greater tumor mass. Overall, if the authors do not add a non-immobilized LC group matching for tumor size with the LC-30 days + 15-days, their data interpretation needs to be changed as to highlight the fact that the bigger tumor burden observed in LC-30 days + 15-days might be what causes the increased wasting observed.

Previous comment: The authors state that a portion of the gastrocnemius muscle was used for histology. Could the authors provide details on which portion they chose (i.e. red vs. white) for which outcomes and how they made sure to consistently sample the same portion for all mice. This is really important given the heterogeneity of the gastrocnemius fiber composition.

Authors’ Response: We thank the reviewer for these comments. As the muscle mass (average 100 mg, depending on the experimental groups) is relatively small, the entire gastrocnemius muscle was used for all the experimental procedures for all the study groups in the current investigation. Thus, no distinction was made between red and white fibers (See lines 938-941 in the revised manuscript). All the fibers were present in the homogenates and used in the histological preparations.

New comment: Unfortunately, lines 938-941 correspond to the bibliography section. I believe the authors were referring to 642-646 instead stating “The gastrocnemius was divided into two halves (cross section) which were identical as to their muscle type composition”. Please clarify whether the halves were cut longitudinally or transversally with respect to the long axis of the muscle and how the authors made sure to trace the same area of the muscle for different animals for CSA and fiber typing measurements. Given the differences in fiber typology across the gastrocnemius muscle, it is primordial that the same muscle area was consistently traced across animals especially given the very low number of fibers traced, which remains insufficient to be truly representative of the gastrocnemius muscle.

Previous comment: Although discussed in the text, the authors to not present quantification of the 20S proteasome C8 nor of the total ubiquitinated protein. These quantifications should be added although I am concerned that total ubiquitinated protein could not be quantified for all samples given the blots.

Authors’ Response: We thank the reviewer for this comment. We regret to say that Figure 4 (panels D & E) in the formerly and revised manuscript versions corresponded, indeed, to measurements of the optical densities and statistical analyses in each experimental group of the markers 20S proteasome and total protein ubiquitination levels. The blots were, indeed, quantified.

New comment: Figure 4 does not include a panel D or E and hence those quantifications do not appear to be depicted in the current or former version of the manuscript.

Reviewer 3 Report

The authors have satisfactorily addressed my original concerns. Some additional minor grammatical editing is still recommended prior to publication.